# Lipid kinases VPS34 and PIKfyve coordinate a phosphoinositide cascade to regulate retriever-mediated recycling on endosomes

Sai Srinivas Panapakkam Giridharan[1†], Guangming Luo[1†], Pilar Rivero-Rios[1], Noah Steinfeld[1], Helene Tronchere[2], Amika Singla[3], Ezra Burstein[3], Daniel D Billadeau[4], Michael A Sutton[5], Lois S Weisman[1]*

[1]Life Sciences Institute and Department of Cellular and Developmental Biology, University of Michigan, Ann Arbor, United States; [2]INSERM U1048 I2MC, France and Université Paul Sabatier, Toulouse, France; [3]Department of Internal Medicine, and Department of Molecular Biology, University of Texas Southwestern Medical Center, Dallas, United States; [4]Division of Oncology Research and Schulze Center for Novel Therapeutics, Mayo Clinic, Rochester, United States; [5]Neuroscience Graduate Program, Molecular and Behavioral Neuroscience Institute, Molecular and Integrative Physiology, University of Michigan, Ann Arbor, United States

*For correspondence: lweisman@umich.edu

†These authors contributed equally to this work

Competing interest: The authors declare that no competing interests exist.

**Abstract** Cell surface receptors control how cells respond to their environment. Many cell surface receptors recycle from endosomes to the plasma membrane via a recently discovered pathway, which includes sorting-nexin SNX17, Retriever, WASH, and CCC complexes. Here, using mammalian cells, we discover that PIKfyve and its upstream PI3-kinase VPS34 positively regulate this pathway. VPS34 produces phosphatidylinositol 3-phosphate (PI3P), which is the substrate for PIKfyve to generate PI3,5P$_2$. We show that PIKfyve controls recycling of cargoes including integrins, receptors that control cell migration. Furthermore, endogenous PIKfyve colocalizes with SNX17, Retriever, WASH, and CCC complexes on endosomes. Importantly, PIKfyve inhibition results in displacement of Retriever and CCC from endosomes. In addition, we show that recruitment of SNX17 is an early step and requires VPS34. These discoveries suggest that VPS34 and PIKfyve coordinate an ordered pathway to regulate recycling from endosomes and suggest how PIKfyve functions in cell migration.

## Editor's evaluation

The authors investigate the role of the PI3P 5-kinase protein (PIKfyve) in endosome to cell surface recycling. They report that PIKfyve function is necessary for cell migration and endosomal recycling of integrin proteins via the SNX17-Retriever pathway. The manuscript will be of broad interest to researchers studying the regulation of integrin trafficking during cell migration, the organization and function of the endosomal network, and the increasingly important area of endosomal sorting de-regulation in a wide array of human diseases.

## Introduction

The functions of many cell surface receptors are controlled in part via the regulation of their exposure to the cell surface. Receptors are removed from the cell surface via regulated endocytosis, and then are either returned via recycling pathways or sent to lysosomes for degradation (*Cullen and Steinberg,*

2018; *Grant and Donaldson, 2009*; *Naslavsky and Caplan, 2018*). Multiple types of receptors are regulated via endocytosis and regulated recycling. These include G-protein-coupled receptors, post-synaptic receptors for neurotransmitters, nutrient transporters, and cell adhesion proteins. Thus, gaining molecular insight into the regulation of endocytosis and mechanisms for receptor recycling is key to understanding the control of multiple physiological processes.

An important endosomal recycling pathway was recently discovered that is regulated in part by the Retriever complex. The Retriever complex is composed of three proteins, VPS29, which is also in the retromer complex, and two unique subunits, VPS35L (C16orf62) and VPS26C (DSCR3). The Retriever complex acts with the sorting nexin, SNX17, the WASH complex, and the CCC complex which includes CCDC22 (coiled-coil domain containing 22)–CCDC93 (coiled-coil domain containing 93), and several of 10 COMMD (copper metabolism MURR1 domain)-containing proteins (*Chen et al., 2019*; *McNally and Cullen, 2018*; *Simonetti and Cullen, 2019*; *Wang et al., 2018*) to regulate trafficking of cargoes and membranes from early endosomes back to the cell surface. Proteomic studies revealed that this pathway traffics over 120 cell surface proteins in HeLa cells (*McNally et al., 2017*), which establishes SNX17-Retriever-CCC-WASH as an important pathway for protein recycling.

The best characterized cargoes of the SNX17-Retriever-CCC-WASH pathway are the integrins, which are transmembrane proteins that control cell migration via regulation of focal adhesion complexes (*Wozniak et al., 2004*). While integrins traffic through the SNX17 pathway, it remains an open question whether integrins can also go to the cell surface via the SNX27 Retromer pathway (*Kvainickas et al., 2016*; *Steinberg et al., 2013*). Note that the levels of Retriever-specific subunits are much lower than Retromer subunits (*Itzhak et al., 2016*).

Integrins that are exposed on the cell surface connect the intracellular actin network to the extracellular matrix (*Vicente-Manzanares et al., 2009*). Integrin levels at the cell surface are controlled both by their endocytosis into endosomes and their subsequent recycling back to the plasma membrane (*Moreno-Layseca et al., 2019*). Thus, understanding how integrin recycling is controlled is of great interest.

Control of the SNX17-Retriever-CCC-WASH pathway likely occurs in part via SNX17 recognition of cargoes as well as SNX17 association with the other proteins of the transport machinery. SNX17 specifically binds β1-integrin as well as other cargo receptors via direct interaction with an NPxY motif on the cytoplasmic side of the cargo protein (*Böttcher et al., 2012*; *Chandra et al., 2019*; *Jia et al., 2014*; *McNally et al., 2017*; *Steinberg et al., 2012*). In addition, SNX17 binds phosphatidylinositol 3-phosphate (PI3P) in vitro (*Chandra et al., 2019*). The WASH complex is recruited to endosomes via the Retromer (*Harbour et al., 2010*; *Harbour et al., 2012*; *Helfer et al., 2013*; *Jia et al., 2012*; *Zavodszky et al., 2014*), and is also likely recruited by binding to PI3P via the FAM21 subunit (*Jia et al., 2010*).

PI3P is one of seven phosphorylated phosphatidylinositol (PPI), which are low abundance signaling lipids that play multiple essential roles (*De Craene et al., 2017*; *Schink et al., 2016*). The seven PPI species are defined by their phosphorylation status at positions 3, 4, 5 of the inositol ring. The synthesis and turnover of PPIs are spatially and temporally regulated by several lipid kinases and phosphatases. PPI lipids act via the recruitment and control of their downstream effector proteins and orchestrate various cellular events including cell signaling, cytoskeletal organization, membrane trafficking, cell migration, and cell division (*Dickson and Hille, 2019*).

Some of the cellular PI3P serves as a substrate for PIKfyve, the lipid kinase that generates phosphatidylinositol 3,5-bisphosphate (PI3,5P$_2$) (*Hasegawa et al., 2017*; *Ho et al., 2012*; *McCartney et al., 2014a*; *Shisheva, 2012*). PIKfyve also serves as the primary source for cellular pools of phosphatidylinositol 5-phosphate (PI5P), likely via the action of lipid phosphatases on PI3,5P$_2$ (*Zolov et al., 2012*) and possibly by direct synthesis (*Shisheva, 2012*). PIKfyve exists in a protein complex which includes the lipid phosphatase, Fig4 and scaffold protein, Vac14. Both Fig4 and Vac14 positively regulate PIKfyve kinase activity (*de Araujo et al., 2020*; *Dove et al., 2009*; *Ho et al., 2012*; *McCartney et al., 2014a*; *Shisheva, 2012*; *Strunk et al., 2020*). Loss of PIKfyve or its positive regulators causes defects in lysosomal homeostasis and results in enlargement of endosomes and lysosomes in a variety of cell types (*de Araujo et al., 2020*; *Dove et al., 2009*; *Ho et al., 2012*; *McCartney et al., 2014a*; *Shisheva, 2012*).

The PIKfyve pathway is critical for normal function of multiple organs and tissues. Fig4 or Vac14 knockout mice die perinatally and exhibit profound neurodegeneration (*Chow et al., 2007*; *Zhang*

*et al., 2007*). Similarly, mutations in FIG4 and *VAC14* have been linked to neurological diseases including Charcot-Marie-Tooth syndrome and amyotrophic lateral sclerosis. Furthermore, homozygous null mutations in human FIG4 lead to infantile death and impairment of multiple organs (*Lenk et al., 2016*; *McCartney et al., 2014a*). Likewise, a PIKfyve hypomorphic mouse mutant dies perinatally and in addition to neurodegeneration has defects in multiple organs including the heart, lung, and kidneys (*Zolov et al., 2012*). A whole-body knockout of PIKfyve results in very early lethality (*Ikonomov et al., 2011*; *Takasuga et al., 2013*). Together these studies indicate that PIKfyve is required for multiple physiological pathways and functions in multiple cell types and organs.

Recent studies indicate that PIKfyve plays roles in cell migration, cell polarization, and cell invasion (*Cinato et al., 2021*; *Dayam et al., 2017*; *Dupuis-Coronas et al., 2011*; *Oppelt et al., 2014*; *Oppelt et al., 2013*; *Shi and Wang, 2018*). However, a full understanding of the molecular mechanisms whereby PIKfyve regulates cell migration remain to be established. Here, we show that PIKfyve activity is required for cell migration. We discover that this occurs in part through regulation of the surface levels of β1-integrin via PIKfyve control of the SNX17-Retriever-CCC-WASH pathway.

## Results

### PIKfyve positively regulates cell migration

Recent studies using siRNA silencing as well as inhibition of PIKfyve indicate that PIKfyve plays a role in cell migration (*Cinato et al., 2021*; *Oppelt et al., 2014*; *Oppelt et al., 2013*; *Shi and Wang, 2018*). To further investigate, we performed wound healing assays in the presence of two chemically distinct PIKfyve inhibitors, YM201636 and apilimod. In cultured cells, compared to vehicle control, PIKfyve inhibition with apilimod and YM201636 delayed wound healing by 45% and 40%, respectively (*Figure 1A–B*). Importantly, inhibition of PIKfyve did not significantly affect cell viability or proliferation (*Figure 1—figure supplement 1*), indicating that the delay in wound healing reflects a decrease in the ability of the cells to migrate.

We also observed a role for PIKfyve in cell migration in primary cells. Primary neonatal cardiac fibroblasts were analyzed in wound healing assays. Inhibition of PIKfyve with YM201636 impaired wound closure by 21% as compared to DMSO-treated cardiac fibroblasts (*Figure 1—figure supplement 2A-B*). Similar results were observed with apilimod (*Cinato et al., 2021*). In addition to chemical inhibition, we sought genetic evidence, and performed wound healing experiments with primary fibroblasts derived from the hypomorphic PIKfyve$^{\beta\text{-geo}/\beta\text{-geo}}$ mouse mutant, which expresses about 10% of the wild-type levels of PIKfyve and half the levels of PI3,5P$_2$ and PI5P (*Zolov et al., 2012*). PIKfyve$^{\beta\text{-geo}/\beta\text{-geo}}$ fibroblasts showed strong defects in cell migration and exhibited a 30% reduction in wound healing compared to wild-type fibroblasts (*Figure 1—figure supplement 2C-D*).

Cell spreading is important for cell migration and can be observed within an hour of seeding cells. This enabled us to monitor the impact of PIKfyve inhibition within a shorter time frame. We assessed the ability of newly seeded cells to spread on plastic dishes coated with fibronectin, a component of the extracellular matrix. We found that inhibition of PIKfyve for 1 hr reduced cell spreading by 22% as compared to DMSO-treated control cells (*Figure 1—figure supplement 2E-F*).

The above studies, as well as previous studies, relied solely on lowering PIKfyve activity. To further test whether PIKfyve has a direct role in cell migration, we took the converse approach and tested whether elevation of PIKfyve activity promotes cell migration. We identified a hyperactive allele, PIKfyve-KYA, which elevates PI3,5P$_2$ and PI5P 4-fold and 2-fold, respectively (*McCartney et al., 2014b*). Inducible expression of PIKfyve-KYA and wild-type PIKfyve was achieved via engineered Flp-In HEK293T cell lines. Induction of PIKfyve-KYA caused a significant increase in wound healing, by approximately 20%, whereas induction of wild-type PIKfyve resulted in a similar degree of cell migration to that observed in control cells (*Figure 1C–D*). The wound healing of these three cell lines was not different in the absence of doxycycline induction (*Figure 1—figure supplement 2G-H*). Together, these results suggest that PIKfyve activity directly impacts cell migration.

### PIKfyve activity plays a role in controlling surface levels of β1-integrin

That PIKfyve plays a direct role in cell migration, provided an opportunity to determine specific pathways that are directly regulated by PIKfyve. Integrins play a crucial role in cell migration. Thus, we tested whether PIKfyve activity correlates with integrin localization. Integrins are heterodimers of α

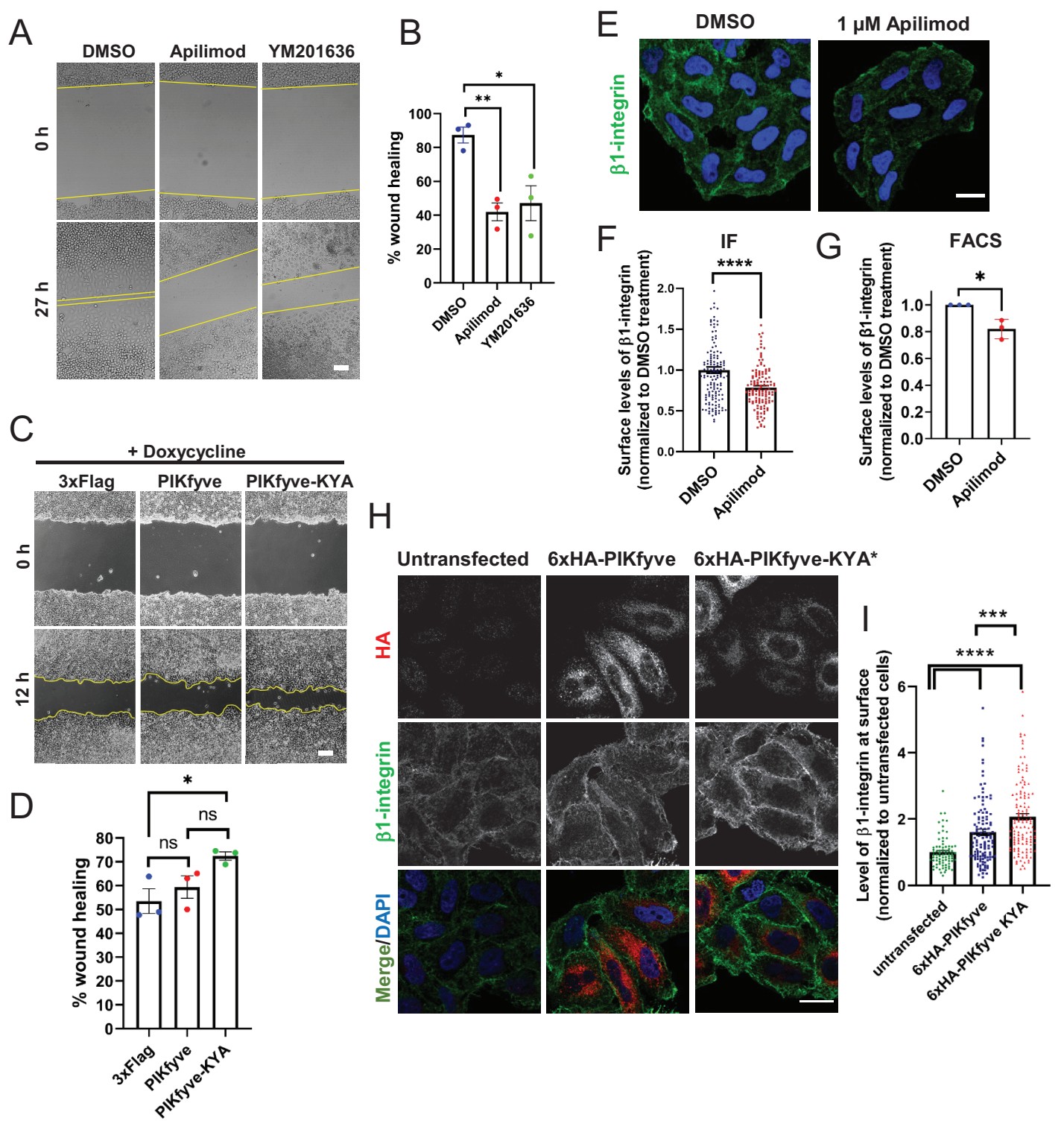

**Figure 1.** PIKfyve positively regulates cell migration in part via regulation of cell surface levels of β1-integrin. (A–B) Inhibition of PIKfyve delays cell migration. (**A**) Wound healing was assessed in HeLa cells following a 27 hr incubation in the presence of either DMSO, 1 μM apilimod, or 0.8 μM YM201636. (**B**) Percentage of wound closure was quantified. Bar: 100 μm. (**C–D**) Increasing PIKfyve activity promotes cell migration. (**C**) Wound healing assays were performed in Flp-in HEK293T cells stably expressing doxycycline-inducible wild-type PIKfyve or hyperactive PIKfyve-KYA in the presence of 100 ng/ml doxycycline for 12 hr. (**D**) Percentage of wound area closure was quantified. Bar: 100 μm. (**E–G**) Inhibition of PIKfyve decreases the surface levels of β1-integrin. (**E**) HeLa cells treated with DMSO or 1 μM apilimod for 1 hr were incubated with antibodies to label surface β1-integrin for 1 hr

*Figure 1 continued on next page*

*Figure 1 continued*

at 4°C and fixed at 4°C. (**F**) Intensity of β1-integrin per cell was quantified and normalized to the average intensity of the DMSO treatment for that particular experiment. Bar: 20 μm. (**G**) HeLa cells treated with DMSO or 1 μM apilimod for 1 hr were incubated with antibodies to label surface β1-integrin for 1 hr at 4°C followed by incubation with 488 Alexa-Fluor-conjugated secondary antibodies for 30 min at 4°C. Cells were fixed and 10,000 cells were analyzed per experiment by flow cytometry. The mean intensity of surface β1-integrin was measured and values normalized to DMSO treatment. (**H–I**) Increasing PIKfyve activity elevates the surface levels of β1-integrin. (**H**) HeLa cells either untransfected or transiently transfected with 6xHA-PIKfyve or 6xHA-PIKfyve-KYA incubated for 1 hr at 4°C with antibodies to label surface β1-integrin. Cells were fixed, permeabilized, immunostained with an anti-HA antibody and corresponding Alexa-Fluor-conjugated secondary antibodies. (**I**) Intensity of β1-integrin per cell was quantified and the values were normalized to the average intensity of untransfected cells for each experiment. Bar: 20 μm. Data presented as mean ± SE. Statistical significance from three independent experiments were determined using unpaired two-tailed Student's t-test (**F**) or paired two-tailed Student's t-test (**G**) or one-way ANOVA and Dunnett's (**B**) or Tukey's (**D,I**) post hoc tests. Yellow lines indicate the migration front. *p < 0.05, **p < 0.01, ***p < 0.005, ****p < 0.001, and ns, not significant.

The online version of this article includes the following source data and figure supplement(s) for figure 1:

**Source data 1.** Contains numerical source data for *Figure 1*.

**Figure supplement 1.** Inhibition of PIKfyve does not affect cell viability or proliferation.

**Figure supplement 1—source data 1.** Contains numerical source data for *Figure 1—figure supplement 1*.

**Figure supplement 2.** PIKfyve is required for cell migration and cell adhesion.

**Figure supplement 2—source data 1.** Contains numerical source data for *Figure 1—figure supplement 2*.

**Figure supplement 3.** Depletion of PIKfyve results in a decrease of the surface levels of β1-integrins.

**Figure supplement 3—source data 1.** Contains numerical and uncropped western blot source data for *Figure 3—figure supplement 1*.

and β chains, and β1-integrin is the most commonly found integrin β subunit (*Moreno-Layseca et al., 2019*). We therefore tested whether PIKfyve activity plays a role in the levels of β1-integrin at the cell surface. Notably, using immunofluorescence or flow cytometry, which allowed us to count many more cells, we found that when compared with DMSO-treated cells, apilimod treatment for 1 hr resulted in 21% and 18% less surface-exposed β1-integrin, respectively (*Figure 1E–G*). To further probe the importance of PIKfyve, we used siRNA to deplete PIKfyve and assessed the impact of loss of PIKfyve on the surface levels of β1-integrin (*Figure 1—figure supplement 3A*). Compared to control siRNA-treated cells, depletion of PIKfyve leads to a decrease in surface levels of β1-integrin by 21%. Importantly, simultaneous expression of siRNA-resistant PIKfyve in siRNA-treated cells rescued integrin levels and furthermore increased them to 21% above the levels in cells treated with control siRNA alone (*Figure 1—figure supplement 3B-C*).

Conversely, we tested whether an increase in PIKfyve activity increases the surface levels of β1-integrin. We transiently expressed wild-type PIKfyve and PIKfyve-KYA in HeLa cells, and observed a 60% and 107% increase, respectively, in β1-integrin on the cell surface as compared to untransfected control cells (*Figure 1H–I*). These findings suggest that PIKfyve activity regulates cell migration in part by regulating surface levels of β1-integrin.

Confidence in the biological significance of the lowered levels of β1-integrin observed by both chemical inhibition and knock-down of PIKfyve is further supported by observations that overexpression of PIKfyve or expression of hyperactive PIKfyve results in an increase in β1-integrin (*Figure 1H–I*). A potential reason why bigger effects due to inhibition or knock-down of PIKfyve are not observed is that some PI3,5$P_2$ and/or PI5P may still be present. In an earlier study we found that fibroblasts from a hypomorphic PIKfyve$^{β-geo/β-geo}$ mouse mutant, which expresses 10% of the wild-type levels of PIKfyve, or shRNA of PIKfyve in wild-type fibroblasts, which reduces PIKfyve to 17% of wild-type levels, each still had approximately half of the normal levels of PI3,5$P_2$ and PI5P (*Zolov et al., 2012*). Similarly, at relatively short time points used for PIKfyve inhibition, there is still some remaining PI3,5$P_2$.

## Inhibition of PIKfyve results in the accumulation of β1-integrin within internal compartments

That the surface levels of β1-integrin decrease following just 1 hr of PIKfyve inhibition suggests that PIKfyve has an acute role in the regulation of integrin levels at the cell surface. To test this further, we determined the fate of surface β1-integrin after acute inhibition of PIKfyve for 15 or 30 min. We labeled HeLa cells with an antibody against an extracellular epitope of β1-integrin and assessed the levels of surface exposed and internalized β1-integrin in the presence of either DMSO or apilimod (*Figure 2*).

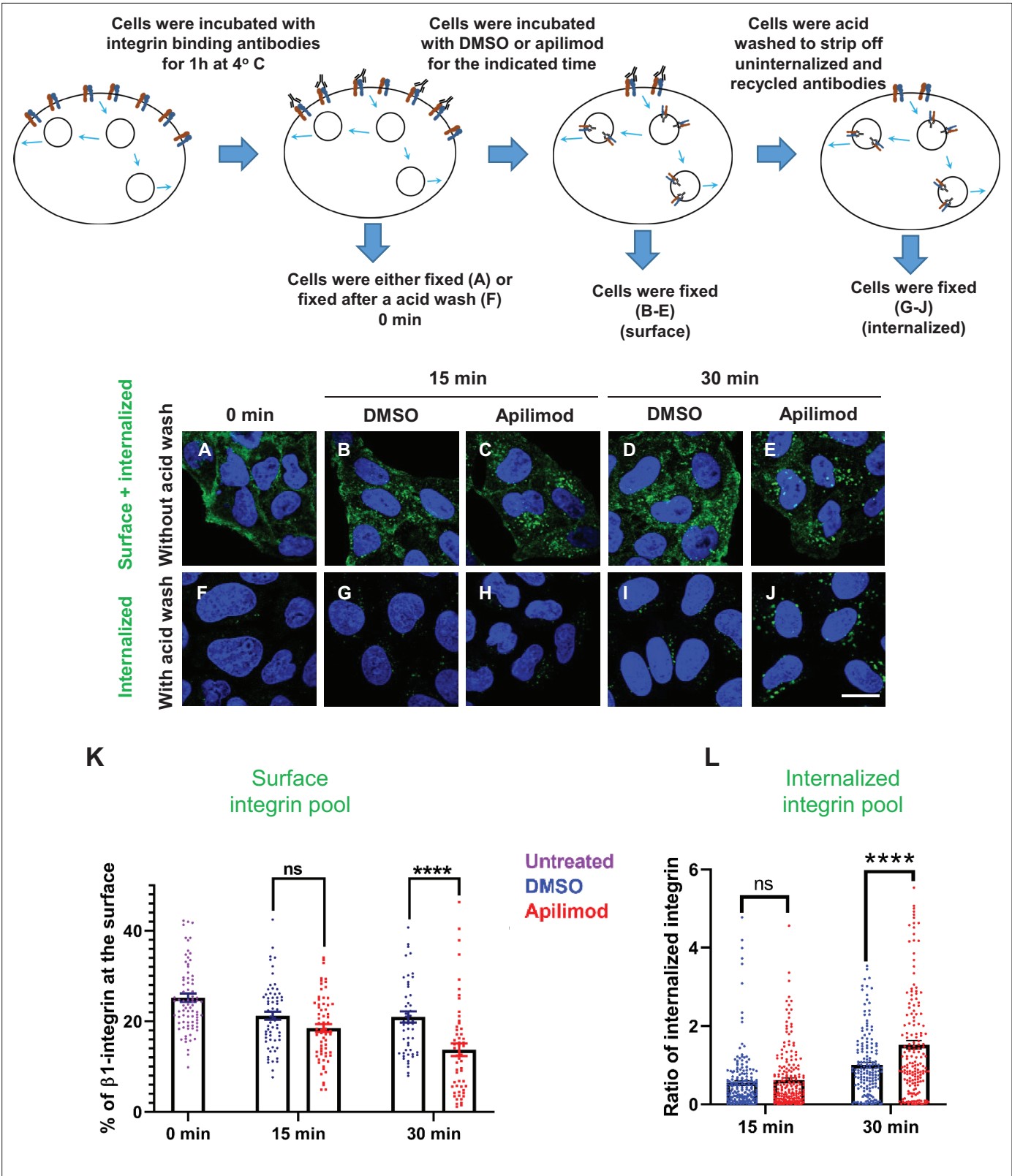

**Figure 2.** Inhibition of PIKfyve causes a rapid loss of β1-integrin from the cell surface and a concomitant accumulation of β1-integrin in internal compartments. (**A–J**) HeLa cells were incubated with antibodies to label surface β1-integrin for 1 hr at 4°C. Cells were either fixed (**A**), fixed after an acid wash (**F**), or incubated with media containing DMSO or 1 µM apilimod at 37°C for the indicated times. Following incubation, cells were fixed (**B–E**) or fixed after an acid wash (**G–J**). Fixed cells were permeabilized and immunostained with Alexa-Fluor-488-conjugated anti-mouse secondary antibodies.

*Figure 2 continued on next page*

*Figure 2 continued*

Flow diagram (top) outlines the experiment. (**K**) The surface levels of β1-integrin were inferred from the intensity of β1-integrin within 0.8 µm from the cell border. Surface β1-integrin (for images A–E) is reported as the percentage of the total labeled β1-integrin. (**L**) Internalized β1-integrin was quantified from cells treated as described in (**G–J**). β1-Integrin intensity was normalized to the average intensity of cells treated with DMSO for 30 min for each experiment. Data presented mean ± SE. Statistical significance from three independent experiments was analyzed using two-way ANOVA and Sidak's multiple comparisons tests. (**K–L**). ****$p < 0.001$ and ns, not significant. Bar: 10 µm.

The online version of this article includes the following source data for figure 2:

**Source data 1.** Contains numerical source data for *Figure 2*.

Using methods to solely label surface integrin, we found that many of the cells were permeabilized during incubation on ice and subsequent fixation. Thus, we used an indirect approach. We fixed and permeabilized the cells, performed immunofluorescence localization of total β1-integrin. To estimate surface β1-integrin, we measured the amount within 0.8 µm of the cell border. In untreated cells, the percent of surface β1-integrin at the cell border was 25%. Following 30 min of incubation in DMSO, the amount of labeled integrin at the cell border exhibited a modest decrease to 21% of total integrin. In contrast, inhibition of PIKfyve for 30 min caused a much larger decrease in the amount of integrin at the cell border to 13.7%. There was also a trend in the decrease in β1-integrin at the cell border following 15 min of apilimod treatment, but this change was not statistically significant (*Figure 2K*).

In parallel, we measured the amount of labeled surface integrin that was internalized. We used a brief acid wash to remove surface β1-integrin-bound antibodies and quantitated the protected, internalized β1-integrin bound to antibody (*Figure 2F–J*). Consistent with the observed changes in surface levels of integrin, we found that after 30 min of apilimod treatment, there was a significant increase of 0.52-fold more internalized β1-integrin as compared to DMSO-treated cells (*Figure 2L*). There was also a trend toward an increase in the internalized pool of β1-integrin after a 15 min treatment, although this change was not statistically significant. These results indicate that PIKfyve has a role in β1-integrin trafficking.

## Endogenous PIKfyve localizes to multiple endosomal compartments

To identify intracellular compartment(s) where PIKfyve may act to regulate the surface levels of β1-integrin, we determined the localization of endogenous PIKfyve. To avoid overexpression-based artifacts, we generated a CRISPR-Cas9-engineered HEK293 cell line that expresses 3xHA-PIKfyve from the endogenous PIKfyve locus. We assessed the colocalization of endogenous 3xHA-PIKfyve with several endo-lysosomal markers including EEA1 (early endosomes), RAB7 (late endosomes), RAB11 (recycling endosomes), LAMP1 and LAMP2 (late endosomes/lysosomes), and VPS35 (Retromer complex). We observed the best colocalization with VPS35 (*Figure 3*). VPS35 associates with endosomes that are active in Retromer-dependent transport (*Gallon and Cullen, 2015*).

There was also a modest colocalization of PIKfyve with RAB7 and EEA1, and some colocalization with RAB11, LAMP1, and LAMP2. The HA antibody uniquely recognized PIKfyve because no signal was observed in non-edited control HEK293 cells (*Figure 3—figure supplement 1*). Localization of PIKfyve to several endosomal compartments fits with our previous studies which found a significant pool of VAC14 on both early and late endosomes (*Zhang et al., 2012*) as well as studies showing that exogenous overexpressed PIKfyve partially colocalizes with early and late endosomes (*McCartney et al., 2014b*; *Rutherford et al., 2006*).

## Inhibition of PIKfyve delays the exit of internalized β1-integrin

β1-integrin cycles between the plasma membrane and endosomes (*Moreno-Layseca et al., 2019*). Since endogenous PIKfyve colocalizes with several endocytic compartments, we specifically tested the impact of PIKfyve inhibition on recycling of internalized β1-integrin back to the cell surface. To generate a pool of labeled, internalized β1-integrin, we performed a pulse, where untreated HeLa cells were incubated for 1 hr in the presence of an antibody against an extracellular epitope of β1-integrin. For the chase, the start of the recycling assay (0 time), cells were exposed to an acid wash to strip off any antibody bound to surface-exposed β1-integrin. Cells were then treated with either apilimod or DMSO and further incubated for 15 or 30 min (*Figure 4A–E*). Notably, at 15 min, the amount of antibody-labeled internalized β1-integrin that recycled back to plasma membrane was significantly lowered by 0.58-fold in cells treated with apilimod as compared to DMSO-treated cells. Moreover,

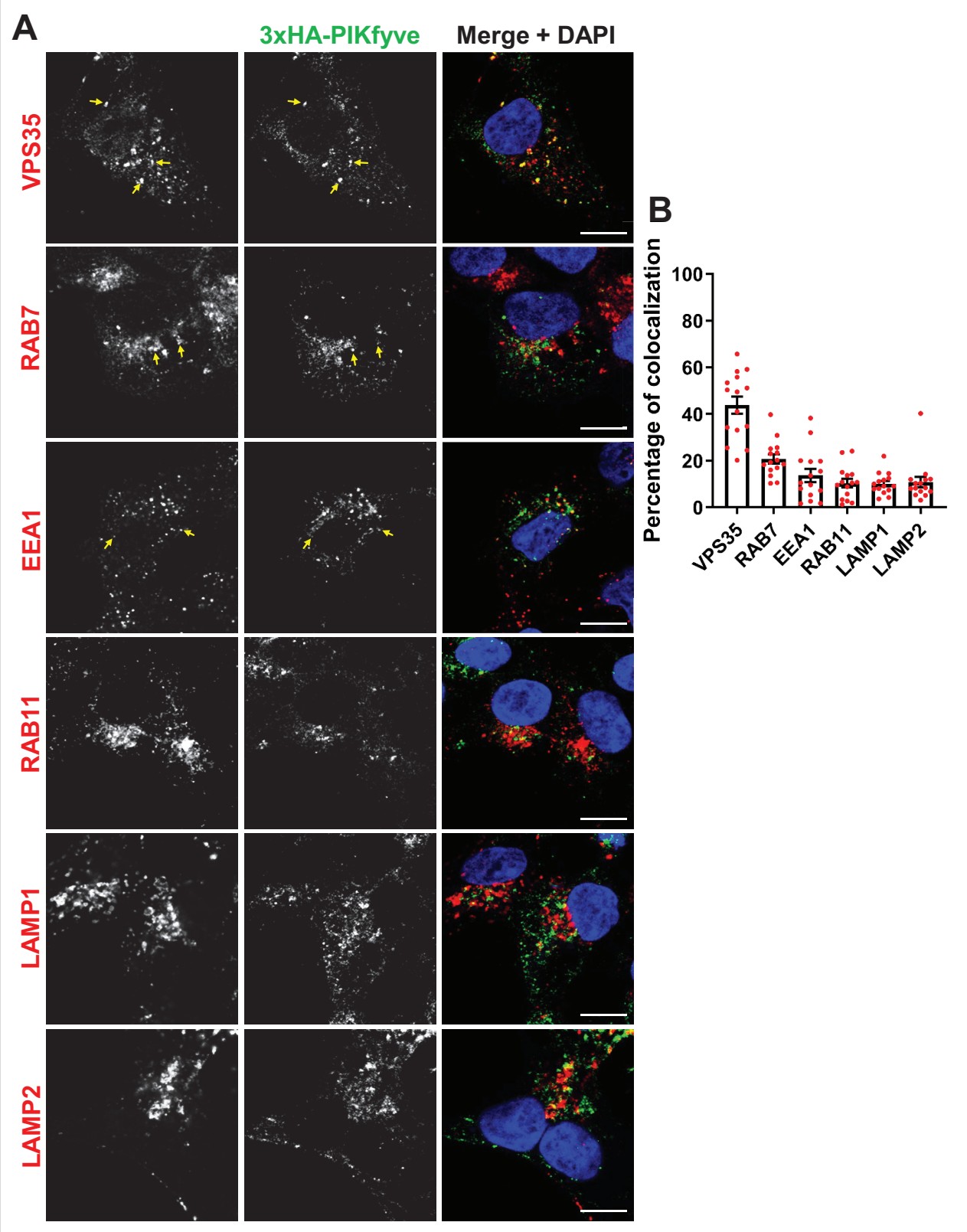

**Figure 3.** PIKfyve resides on early and late endosomes and exhibits the highest colocalization with VPS35 containing endosomes. (**A**) HEK293 cells expressing 3xHA-endogenously tagged PIKfyve were fixed, permeabilized, and immunostained with antibodies against the HA tag and with antibodies against proteins associated with the retromer (VPS35), early endosomes (EEA1), late endosomes (RAB7), recycling endosomes (RAB11), or lysosomes (LAMP1 and LAMP2). Arrows indicate examples of puncta showing colocalization. Bar: 10 μm. (**B**) The percentage of PIKfyve that colocalizes with the

*Figure 3 continued on next page*

*Figure 3 continued*

indicated proteins was determined using Mander's colocalization coefficient from three independent experiments.

The online version of this article includes the following source data and figure supplement(s) for figure 3:

**Source data 1.** Contains numerical source data for *Figure 3*.

**Figure supplement 1.** Immunofluorescence localization of endosomal proteins in unedited HEK293 cells (control for *Figure 3*).

after 30 min of treatment, the apilimod-treated cells had 0.45-fold less surface β1-integrin compared with DMSO-treated cells (*Figure 4K*). In addition, we determined the internalized pool of β1-integrin that remained trapped in internal compartments. At the 15 and 30 min time points, we performed a second acid wash to remove antibodies attached to integrin that returned back to the cell surface (*Figure 4F–J*). Consistent with a defect in the return of β1-integrin back to the cell surface, inhibition of PIKfyve resulted in an increase in the internal, labeled, non-recycled pool of β1-integrin. In cells treated with apilimod for 15 min, the internal-non-recycled pool of β1-integrin was significantly higher by 20% compared with DMSO-treated cells, and this accumulation was 30% higher after 30 min of apilimod treatment (*Figure 4L*). These results indicate that PIKfyve is required for the recycling of β1-integrin from internal compartments to the plasma membrane.

## Inhibition of PIKfyve delays the exit of integrin from several endocytic compartments

β1-integrin traffics through several endocytic compartments. Following internalization into endosomes, most of β1-integrin is recycled back to the plasma membrane either by a slow (RAB11) or fast (RAB4) recycling pathway. To determine whether inhibition of PIKfyve delays the exit of β1-integrin from a specific type of endosome, we first generated a pool of labeled, internalized β1-integrin. HeLa cells were incubated for 1 hr with antibodies that bind to surface-exposed β1-integrin to label the fraction that was internalized during this time frame. Then the remaining surface-bound uninternalized antibodies were removed with a short acid wash (0 min, untreated). Consistent with previous studies, at 0 time, β1-integrin was predominantly present in EEA1-positive endosomes (untreated cells, *Figure 5A–B*). In addition, there was some colocalization of β1-integrin with RAB11-, RAB4-, and LAMP1-positive compartments.

To test whether PIKfyve inhibition altered the exit of internalized β1-integrin out of these compartments, cells were incubated with DMSO or apilimod for the indicated time points. Following treatment, an acid wash was performed to remove any antibody-bound integrin that was recycled back to the cell surface, and the remaining internalized pool was assessed. When compared with DMSO controls, PIKfyve inhibition resulted in significantly more labeled β1-integrin in each of the compartments that are part of its itinerary (*Figure 5B*). This suggests that PIKfyve activity is required for the exit of β1-integrin from several endocytic compartments.

Specifically, the amount of integrin present in EEA1 compartments in DMSO-treated cells decreased by 35% at 15 min and this amount further decreased to 56% at 30 min. In contrast, with PIKfyve inhibition, there was not a significant decrease in the amount of integrin present in EEA1 compartments at either 15 or 30 min. This indicates a delay in the exit of integrin from early endosomes. Similarly, in DMSO-treated cells, the amount of integrin present in LAMP1-positive compartments decreased approximately 32% at 15 and 30 min. However, following PIKfyve inhibition, there was no observable decrease in integrin in LAMP1 compartments. Together, these data show that PIKfyve inhibition causes a delay in the exit of β1-integrin from early and late endosomes. Note that, β1-integrin is recycled from both of these compartments (*Moreno-Layseca et al., 2019*).

Some of the β1-integrin in LAMP1 compartments could potentially be targeted for lysosomal degradation. However, at the time points measured, we did not see an impact of PIKfyve inhibition on the degradation of β1-integrin. The total level of β1-integrin was not significantly altered after apilimod treatment for 30 min (*Figure 5—figure supplement 1*).

We also tested whether PIKfyve activity is required for the trafficking of β1-integrin through either RAB4 or RAB11 compartments, which are part of the fast and slow recycling pathways, respectively. In cells treated with DMSO for 15 or 30 min, the amount of β1-integrin in RAB11-positive compartments decreased by approximately 66%. In contrast, during PIKfyve inhibition the amount of integrin in apilimod-treated cells had a more modest decrease of 22% and 34%, at 15 and 30 min of treatment,

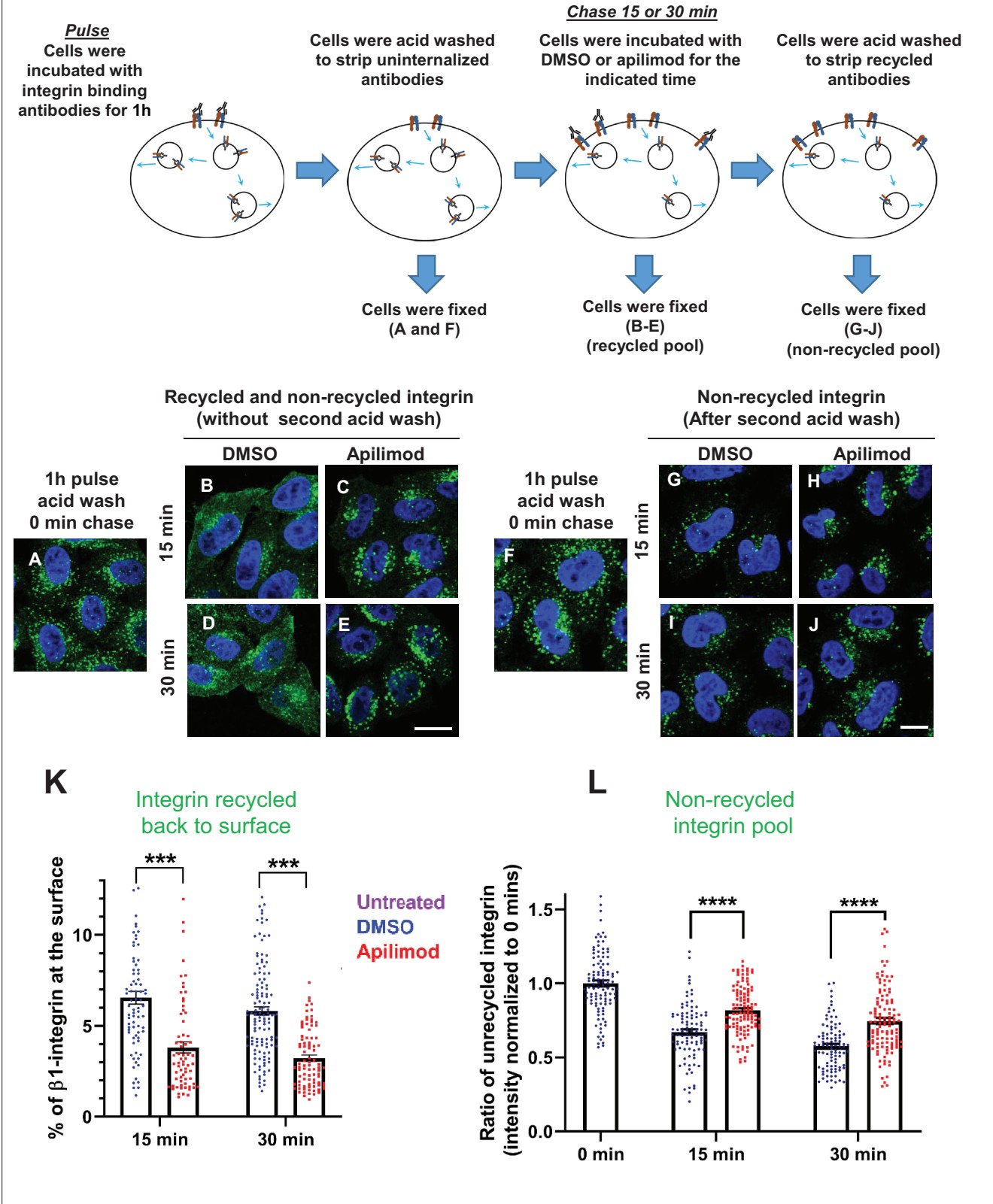

**Figure 4.** Inhibition of PIKfyve results in a defect in β1-integrin recycling. (**A–J**) HeLa cells were incubated with β1-integrin antibody for 1 hr at 37°C to allow the antibody-labeled integrin to internalize. Cells were then acid washed to remove surface β1-integrin-bound antibodies, and were either fixed (**A and F**) or incubated with DMSO or 1 μM apilimod containing media at 37°C for the indicated times. Cells were either fixed (**B–E**) or fixed after a second acid wash to remove antibodies that returned to the surface (**G–J**). Fixed cells were permeabilized and immunostained with Alexa-Fluor-488-conjugated

*Figure 4 continued on next page*

*Figure 4 continued*

anti-mouse secondary antibodies. Flow diagram (top) outlines the experiment. (**K**) Surface levels of β1-integrin were inferred from the intensity of β1-integrin within 0.8 μm from the cell border. The levels of β1-integrin that recycled back to the surface (for images B–E) were quantified as percentage of the total labeled integrin. (**L**) Intensity of non-recycled β1-integrin was quantified from cells treated as indicated in (**G–J**). All values were normalized to the average of the 0 min time point (**F**). Data presented as mean ± SE. Statistical significance from three independent experiments were analyzed using two-way ANOVA and Sidak's multiple comparisons tests. (**K–L**). ***$p < 0.005$ and ****$p < 0.001$. Bar: 10 μm.

The online version of this article includes the following source data for figure 4:

**Source data 1.** Contains numerical source data for *Figure 4*.

respectively. The difference between DMSO and apilimod-treated cells suggested either a decrease in the rate of exit of integrin from RAB11 compartments or increased transport of integrin to RAB11 compartments from early endosomes. However, since the exit of integrin from early endosomes is also defective, the increase in integrin in RAB11 endosomes is likely due to defects in recycling of β1-integrin toward the plasma membrane.

There were similar defects in the trafficking of β1-integrin from RAB4 compartments. In cells treated with DMSO for 15 and 30 min, the amount of integrin remaining in RAB4 endosomes was significantly lower by 57% and 67%, respectively. In comparison, following PIKfyve inhibition, the decrease in β1-integrin in RAB4 endosomes was only 27% at 15 min and 48% after 30 min. Thus, short-term inhibition of PIKfyve also slows the recycling of β1-integrin from RAB4 endosomes. Together, these studies suggest that PIKfyve plays a role in the recycling of β1-integrin from all endocytic compartments tested including early and late endosomes as well as fast and slow recycling endosomes.

## PIKfyve colocalizes with SNX17-Retriever-CCC-WASH complex proteins

β1-integrin recycling from endosomes is regulated by the sorting nexin, SNX17, the Retriever complex, the CCC complex, and the actin regulatory WASH complex (*Chen et al., 2019*; *McNally and Cullen, 2018*; *Simonetti and Cullen, 2019*; *Wang et al., 2018*). Loss of SNX17 inhibits the recycling of β1-integrin from endosomes (*Böttcher et al., 2012*; *McNally et al., 2017*; *Steinberg et al., 2012*).

β1-integrin is one of several cargoes that require SNX17, WASH, Retriever, and the CCC complex. To test whether PIKfyve is more generally required for SNX17, Retriever, CCC, and WASH complex-mediated trafficking from endosomes to the plasma membrane, we tested the impact of PIKfyve inhibition on two additional Retriever cargoes, α5-integrin and low-density lipoprotein receptor-related protein 1 (LRP1) (*Farfán et al., 2013*; *McNally et al., 2017*). We performed surface biotinylation assays and found that inhibition of PIKfyve lowers β1-integrin, α5-integrin, and LRP1 levels on the cell surface by approximately 50% each (*Figure 6A–B*). As an orthogonal approach, we tested the changes in α5-integrin following depletion of PIKfyve by siRNA (*Figure 6—figure supplement 1*). Similar to PIKfyve inhibition, depletion of PIKfyve also caused a decrease in surface levels of α5-integrin levels by 17%. PIKfyve re-expression led to increase in surface levels by 47% more than the control-treated cells. Together, these findings suggest that PIKfyve regulates general SNX17-Retriever-CCC-WASH-mediated recycling.

To test if depletion of PIKfyve non-specifically affects all receptors on surface, we tested the effect of PIKfyve inhibition on surface levels of EGFR. We chose EGFR because in a previous study we found that its trafficking was not affected in mouse embryonic fibroblast (MEF) generated from Vac14-/-mice (*Zhang et al., 2007*). Notably, inhibition of PIKfyve using conditions that impacted integrins and LRP1 does not alter the surface levels of EGFR (*Figure 6—figure supplement 2*). These findings suggest that acute inhibition of PIKfyve does not impact all membrane trafficking pathways that require endosomal function.

In further support that PIKfyve is required for the SNX17-Retriever-CCC-WASH pathway, we tested and found that PIKfyve colocalizes with SNX17 and subunits of the WASH (Strumpellin and FAM21), Retriever (VPS35L), and CCC (COMMD1 and CCDC93) complexes (*Figure 6C–D*). Utilizing Mander's coefficient, we quantified the fraction of the indicated proteins that overlap with endogenous PIKfyve-positive puncta and observed a colocalization of 30–50% of endogenous PIKfyve with the proteins implicated in β1-integrin recycling. This colocalization was not observed in the non-edited control HEK293 cells (*Figure 6—figure supplement 3*). These data provide further support for the hypothesis that PIKfyve regulates β1-integrin recycling from endosomes via regulation of the SNX17-Retriever-CCC-WASH complex.

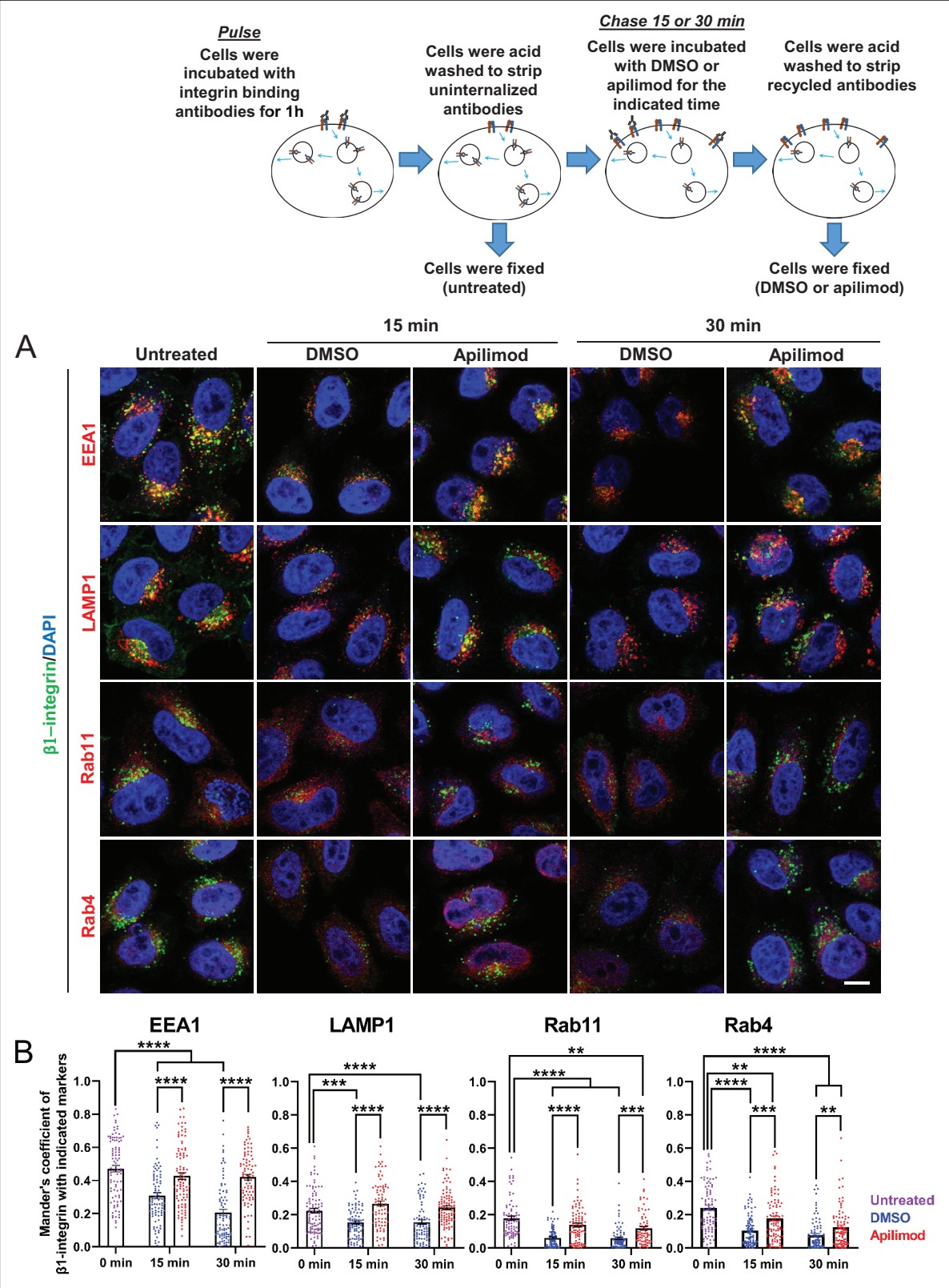

**Figure 5.** Inhibition of PIKfyve results in a defect in β1-integrin recycling from late endosomes, early endosomes, and recycling endosomes. (**A**) HeLa cells were incubated with β1-integrin antibody for 1 hr at 37°C to allow antibody-labeled surface integrin to internalize and then acid washed to remove surface antibodies. Cells were either fixed (0 min, untreated) or treated with either DMSO or apilimod for the indicated time points. Cells were then acid washed, fixed, and the localization of internalized β1-integrin was analyzed with well-established marker proteins: EEA1, LAMP1, RAB11, and

*Figure 5 continued on next page*

*Figure 5 continued*

RAB4. (**B**) Colocalization of the internalized β1-integrin pool with endocytic markers was determined using Mander's colocalization coefficient analysis. Data presented as mean ± SE. Statistical significance from three independent experiments was analyzed using two-way ANOVA and Sidak's multiple comparisons tests. **p < 0.01, ***p < 0.005, and ****p < 0.001. Bar: 10 µm.

The online version of this article includes the following source data and figure supplement(s) for figure 5:

**Source data 1.** Contains numerical source data for *Figure 5*.

**Figure supplement 1.** β1-Integrin levels remain stable during PIKfyve inhibition.

**Figure supplement 1—source data 1.** Contains numerical and uncropped western blot source data for *Figure 5—figure supplement 1*.

## PIKfyve regulates the localization of the CCC and Retriever complexes at endosomes

To gain mechanistic insight into how PIKfyve regulates β1-integrin recycling, we used HeLa cells and tested whether PIKfyve is required for the recruitment of SNX17 and/or Retriever-CCC-WASH complex subunits to endosomes. We tested colocalization of these proteins with VPS35-positive endosomes because both the Retromer and Retriever pathways emerge from VPS35 containing endosomes (*McNally et al., 2017*; *Singla et al., 2019*). In addition, PIKfyve exhibits a strong colocalization with VPS35 (*Figure 3*). Importantly, for each CCC and Retriever subunit tested, acute inhibition of PIKfyve for 30 min caused a significant decrease in their colocalization with VPS35 endosomes. The CCC subunits COMMD1, COMMD5, and CCDC93 were lowered by 22%, 20%, and 30%, respectively (*Figure 7*, *Figure 7—figure supplement 1*). Note that the loss of the CCC proteins from VPS35 endosomes occurred over a relatively short time frame, 30 min, which suggests that PIKfyve plays a direct role. As an orthogonal approach, we tested the changes in COMMD1 localization during depletion of PIKfyve and observed a similar trend. Compared to mock siRNA-treated cells, siRNA depletion of PIKfyve resulted in a 15% decrease in endosomal localization of COMMD1. Importantly this decrease was rescued by expression of PIKfyve (*Figure 7—figure supplement 2*). That the CCC subunits tested partially rely on PIKfyve for their localization suggests that some proteins in the CCC complex may directly bind $PI3,5P_2$ and/or PI5P. Notably, COMMD1, COMMD7, and COMMD10 bind some phosphoinositides including $PI3,5P_2$ and in some cases PI5P in in vitro assays (*Healy et al., 2018*).

We also found that the Retriever subunit, VPS35L, was lowered by 17% (*Figure 7A–B*). We were unable to test VPS26C, the other subunit unique to Retriever, because we have not identified an antibody suitable for immunofluorescence. The reliance of the Retriever complex on PIKfyve may either be due to direct binding of some Retriever subunits to $PI3,5P_2$ and/or PI5P, or may be due to a requirement for the presence of the CCC complex on endosomes, since the Retriever complex interacts with the CCC complex, and a portion of VPS35L is associated with the CCC complex (*McNally et al., 2017*; *Phillips-Krawczak et al., 2015*; *Singla et al., 2019*).

In contrast with the partial loss of CCC and Retriever proteins from membranes, acute inhibition of PIKfyve resulted in a 27% increase in SNX17 on VPS35-positive endosomes (*Figure 7A–B*). The increased recruitment of SNX17 may be due to an elevation in PI3P that occurs during inhibition of PIKfyve (*Zolov et al., 2012*). The WASH complex subunit, FAM21, remained unchanged.

To further determine whether the effects observed during PIKfyve inhibition were due to elevation in PI3P or lowering $PI3,5P_2$ or PI5P, we sought to determine the enzyme that generates the PI3P pool that recruits SNX17. In MEFs, approximately, two-thirds of the PI3P pool comes from VPS34 (*Devereaux et al., 2013*; *Ikonomov et al., 2015*). Note that inhibition of VPS34 also lowers $PI3,5P_2$ and PI5P (*Figure 7—figure supplement 3*; *Devereaux et al., 2013*; *Ikonomov et al., 2015*). This is because $PI3,5P_2$ is synthesized from PI3P (*Sbrissa et al., 1999*), and PI5P is synthesized from $PI3,5P_2$ (*Zolov et al., 2012*).

To further test a role for PI3P in the recruitment of SNX17, we inhibited VPS34 with VPS34-IN1, and found that there was significantly less SNX17 on VPS35 endosomes (*Figure 7—figure supplement 4*). In addition, we observed less VPS35L and COMMD1. These changes are likely due to lower levels of $PI3,5P_2$ and PI5P, as inhibition of PIKfyve with apilimod also lowered the levels of these proteins on VPS35 endosomes. The levels of FAM21 were also lower. These changes are likely due to lower levels of the PI3P, $PI3,5P_2$, and/or PI5P.

To determine effects due to lowering $PI3,5P_2$ and PI5P, under conditions where PI3P levels are not elevated, we tested and found that a combination of 1 µM apilimod, and 0.1 µM VPS34-IN1, resulted

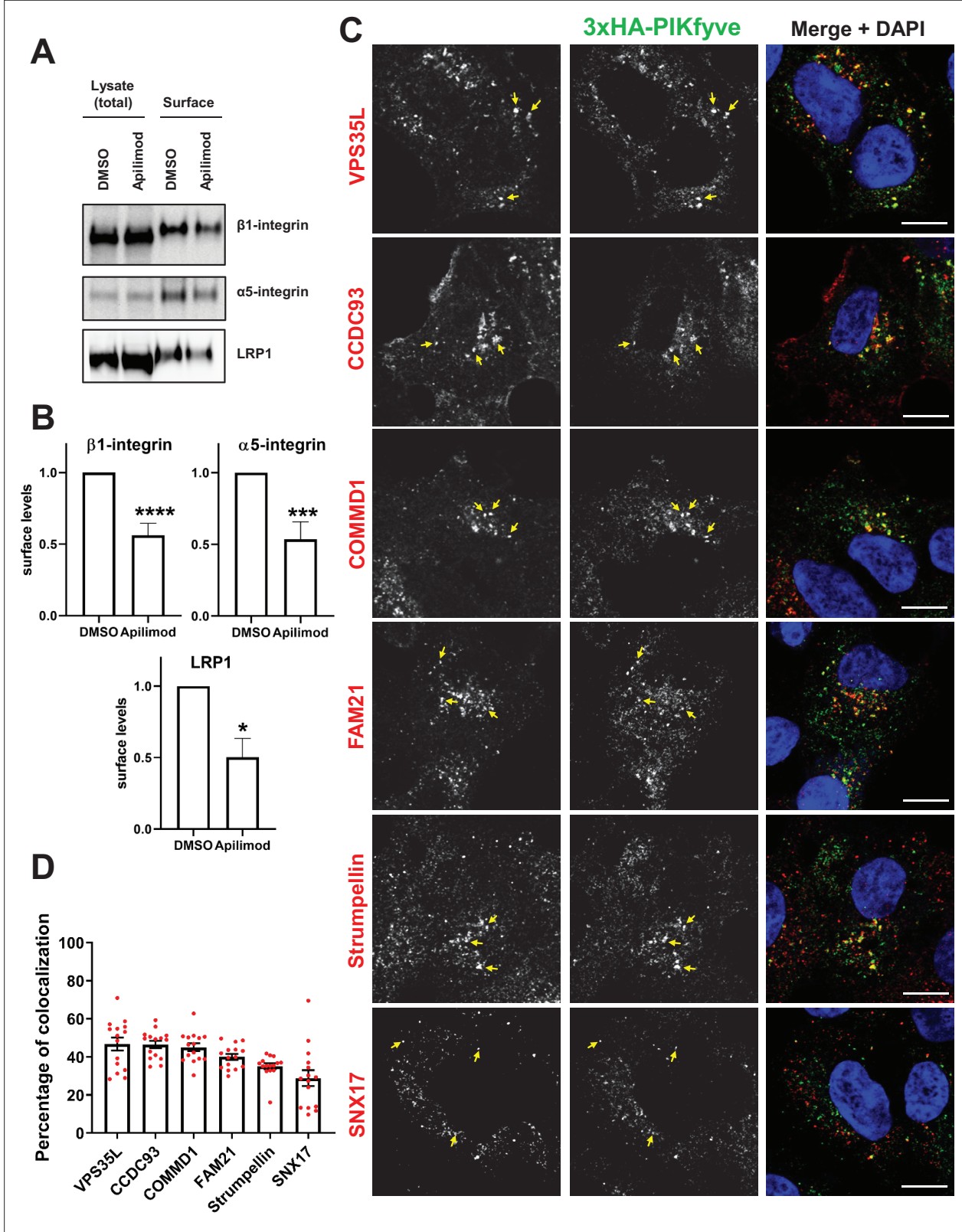

**Figure 6.** PIKfyve regulates the recycling of SNX17-Retriever-CCC-WASH cargoes and colocalizes with SNX17 and subunits of the Retriever, CCC, and WASH complexes. (**A**) HeLa cells were treated with DMSO or 1 μM apilimod for 1 hr and then the levels of SNX17 cargoes, β1-integrin, and α5-integrin were determined using a surface biotinylation assay. Surface biotinylation assay was similarly performed on HUH7 cells and low-density lipoprotein receptor-related protein 1 (LRP1) levels were measured. (**B**) Quantification of western blots from three independent surface biotinylation experiments.

*Figure 6 continued on next page*

*Figure 6 continued*

(**C**) HEK293 cells expressing 3xHA-endogenously tagged PIKfyve were fixed, permeabilized, and incubated with antibodies against the HA tag and antibodies against either SNX17, the Retriever complex subunit (VPS35L), CCC complex subunits (CCDC93 and COMMD1), or WASH complex subunits (FAM21 and Strumpellin). Bar: 10 μm. Arrows indicate examples of puncta showing colocalization. (**D**) The percentage of PIKfyve colocalizing with the indicated proteins was determined using Mander's colocalization coefficient analysis from three independent experiments. Data presented as mean ± SE. *p < 0.05, ***p < 0.005, and ****p < 0.001.

The online version of this article includes the following source data and figure supplement(s) for figure 6:

**Source data 1.** Contains numerical and uncropped western blot source data for *Figure 6*.

**Figure supplement 1.** Depletion of PIKfyve results in a decrease of the surface levels of α5-integrins.

**Figure supplement 1—source data 1.** Contains numerical source data for *Figure 6—figure supplement 1*.

**Figure supplement 2.** Inhibition of PIKfyve does not alter the surface levels of EGFR.

**Figure supplement 2—source data 1.** Contains numerical source data for *Figure 1—figure supplement 2*.

**Figure supplement 3.** Immunofluorescence localization of endosomal proteins in unedited HEK293 cells (control for *Figure 6*).

in lower PI3,5P$_2$ and PI5P, but normal cellular levels of PI3P (*Figure 7—figure supplement 5*). Note that the methods used here to measure phosphoinositides, report on total cellular levels, and would not detect a change at a specific membrane domain.

We tested and found that lowering PI3,5P$_2$ and PI5P under conditions without detectable changes in PI3P resulted in no change in SNX17 recruitment. However, there was less VPS35L, COMMD1, and FAM21 on VPS35 endosomes. We tested whether we could use an orthogonal biochemical approach, mild digitonin treatment, to assess changes in membrane binding of the SNX17 pathway proteins. However, this treatment removed 80–95% of many of the tested subunits. This indicates that most of the complex is either weakly associated with membranes or cytosolic. It is not possible to obtain good quantitative data of an additional 15–20% lowering of this small amount of remaining protein. Thus, this approach is not useful for measuring changes in the membrane pool (*Figure 7—figure supplement 6*).

A summary of trends in phosphoinositide levels with each treatment are shown in *Figure 7—figure supplement 7*. These studies indicate that conversion of PI3P to PI3,5P$_2$ or PI5P plays an important role in ordered recruitment of these complexes. While SNX17 requires PI3P to associate with these endosomes, VPS35L (Retriever) and COMMD1 (CCC) require the synthesis of PI3,5P$_2$ and/or PI5P. Additionally, the localization of FAM21 to endosomes requires the generation of either PI3P, PI3,5P$_2$, and/or PI5P. Consistent with this finding, FAM21 binds multiple phosphorylated phosphoinositide lipids in vitro, including PI3P (*Jia et al., 2010*). Together, these findings suggest that that the SNX17-Retriever-CCC-WASH recycling pathway may be ordered by a phosphoinositide cascade, where PI3P is necessary for the recruitment of SNX17 to WASH complex-containing endosomes, and PI3P-dependent recruitment of PIKfyve generates PI3,5P$_2$ and/or PI5P which facilitates the recruitment the CCC and Retriever complexes.

To further probe how PIKfyve recruits the CCC complex, we focused on COMMD1, a CCC subunit which functions as an obligate dimer and binds multiple phosphorylated phosphoinositide lipids including PI3,5P$_2$ and PI5P in vitro (*Healy et al., 2018*). Mutation of residues that comprise a basic patch on COMMD1, R133Q, H134A, and K167A (COMMD1-QAA) abolished the ability of COMMD1 to bind phosphoinositide lipids in vitro (*Healy et al., 2018*). However, this mutant bounds membranes in cells, although there were no further tests of function. Note that in this mutant, two of the basic residues were substituted with alanine, which is hydrophobic and could potentially cause non-specific sticking to cellular membranes. Thus, we mutated the same sites to R133E, H134Q, and K167E (COMMD1-EQE). We expressed this mutant as well as the original COMMD1-QAA mutant in COMMD1-/- cells and found that as previously reported, there was no statistically significant difference in the binding of COMMD1 and the original COMMD1-QAA mutant to endosomes, 17.1% and 15.9%, respectively. In contrast, the COMMD1-EQE mutant exhibited a statistically significant defect in its association with VPS35 endosomes, where 14.8% bound (*Figure 8B–C*, *Figure 8—figure supplement 1*). These findings suggest that the phosphoinositide binding site of COMMD1 contributes to its association with membranes.

To determine the functional significance of COMMD1 binding to phosphoinositide lipids for β1-integrin recycling, we expressed wild-type COMMD1, COMMD1-QAA, and COMMD1-EQE mutants

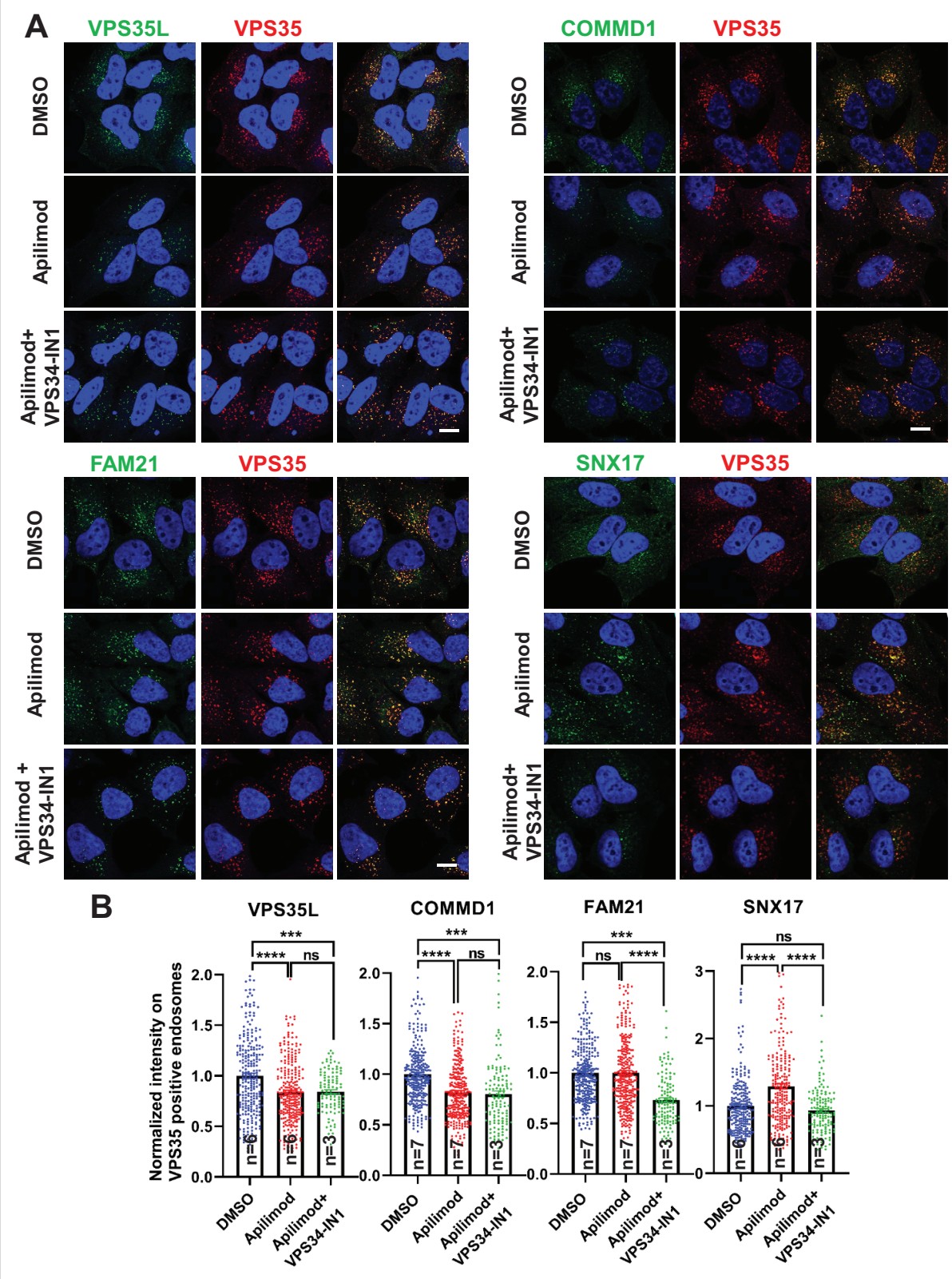

**Figure 7.** CCC and Retriever complexes require PI3,5P$_2$ and/or phosphatidylinositol 5-phosphate (PI5P) to bind to endosomes. (**A**) HeLa cells treated with either DMSO or 1 µM apilimod or co-treated with 1 µM apilimod and 0.01 µM VPS34-IN1 for 30 min were fixed, permeabilized and co-stained with antibodies against VPS35 (**A–D**) and antibodies against either VPS35L, COMMD1, FAM21, or SNX17. (**B**) A mask of VPS35-positive endosomes was generated, and the intensity of VPS35L, COMMD1, FAM21, and SNX17 within this location was quantified. Values were normalized to the corresponding

*Figure 7 continued on next page*

*Figure 7 continued*

average intensity of the DMSO treatment cohort. Data presented as mean ± SE. Statistical significance from three or more independent experiments as indicated within bar graph were analyzed using one-way ANOVA and Tukey's post hoc tests. ***$p < 0.005$ and ****$p < 0.001$, and ns, not significant. Bar: 10 μm.

The online version of this article includes the following source data and figure supplement(s) for figure 7:

**Source data 1.** Contains numerical source data for *Figure 7*.

**Figure supplement 1.** Acute inhibition of PIKfyve results in a loss of the CCC complex subunits, COMMD5 and CCDC93 from endosomes.

**Figure supplement 1—source data 1.** Contains numerical source data for *Figure 7—figure supplement 1*.

**Figure supplement 2.** Depletion of PIKfyve causes a loss of the endosomal localization of COMMD1 which is rescued by PIKfyve expression.

**Figure supplement 2—source data 1.** Contains numerical source data for *Figure 7—figure supplement 2*.

**Figure supplement 3.** Apilimod and VPS34-IN1 are potent inhibitors of PIKfyve and VPS34, respectively.

**Figure supplement 3—source data 1.** Contains numerical source data for *Figure 7—figure supplement 3*.

**Figure supplement 4.** Inhibition of the PI3-kinase, VPS34 results in a loss of SNX17, CCC, Retriever, and WASH complexes from endosomes.

**Figure supplement 4—source data 1.** Contains numerical source data for *Figure 7—figure supplement 4*.

**Figure supplement 5.** Partial inhibition of VPS34 with 0.1 mM VPS34-IN1 combined with treatment with apilimod prevents the elevation of total cellular pools of phosphatidylinositol 3-phosphate (PI3P).

**Figure supplement 5—source data 1.** Contains numerical source data for *Figure 7—figure supplement 5*.

**Figure supplement 6.** Mild cell permeabilization resulted in the release of SNX17-related proteins.

**Figure supplement 6—source data 1.** Contains numerical source data and uncropped western blot source data for *Figure 7—figure supplement 6*.

**Figure supplement 7.** Summary of the effect of inhibitor treatment on phosphoinositide lipid levels and the endosomal localization of SNX17, WASH, CCC, and Retriever subunits.

**Figure supplement 8.** Acute inhibition of PIKfyve does not alter cortactin colocalization on Vps35 endosomes.

**Figure supplement 8—source data 1.** Contains numerical source data for *Figure 7—figure supplement 8*.

in COMMD1-/- cells. Cells were incubated with antibodies against β1-integrin for 1 hr to allow the antibodies to internalize, then the remaining surface-bound antibodies were removed with an acid wash. Immunofluorescence localization of the internalized pool of β1-integrin revealed no significant difference in β1-integrin internalization in untransfected cells or cells transfected with wild-type COMMD1 or the COMMD1-QAA or COMMD1-EQE mutants (*Figure 9A–B*). That there was no difference allowed us to further test recycling of the internalized β1-integrin pool. Cells with internalized β1-integrin were then incubated in serum containing media for 1 hr and the non-recycled pool of β1-integrin was assessed following a second acid wash (*Figure 9C–D*). Cells expressing either wild-type COMMD1 or COMMD1-QAA mutant exhibited a partial rescue of the recycling defect observed in non-transfected COMMD1-/- cells, and retained 23% less non-recycled β1-integrin compared to untransfected cells. Importantly, the COMMD-EQE mutant failed to rescue the recycling defect. Cells expressing the COMMD1-EQE mutant retained 36.2% more integrin than cells rescued with COMMD1 or COMMD1-QAA, respectively. It was surprising that the COMMD1-QAA and COMMD1-EQE mutant behaved differently in their ability to recycle β1-integrin, since each carried amino acid substitutions at the same three residues.

To probe this further, we tested the relative expression of wild-type COMMD1 and the COMMD1-QAA or COMMD1-EQE mutants. Surprisingly, we found that the COMMD1-QAA mutant was expressed at 8.6-fold higher levels than wild-type COMMD1. This much higher level of expression may explain why the COMMD1-QAA mutant rescued recycling and bound normally to membranes, while the COMMD1-EQE exhibited a defect.

The assumption was that the surface residues, R133, H134, and K167, are solely involved in binding to phosphoinositide lipids. However, we tested and found that mutation of these sites also resulted in a partial defect in COMMD1 association with the CCC subunit, CCDC93. We expressed wild-type COMMD1 and the COMMD1-QAA or COMMD1-EQE mutants in HeLa cells and found that mutating R133, H134, and K167 partially affected the binding between CCDC93 and COMMD1 (*Figure 9— figure supplement 1*). When accounting for the amount of COMMD1 pulled down in each experiment, both the COMMD1-QAA and COMMD1-EQE mutant had 40% and 60% less association with

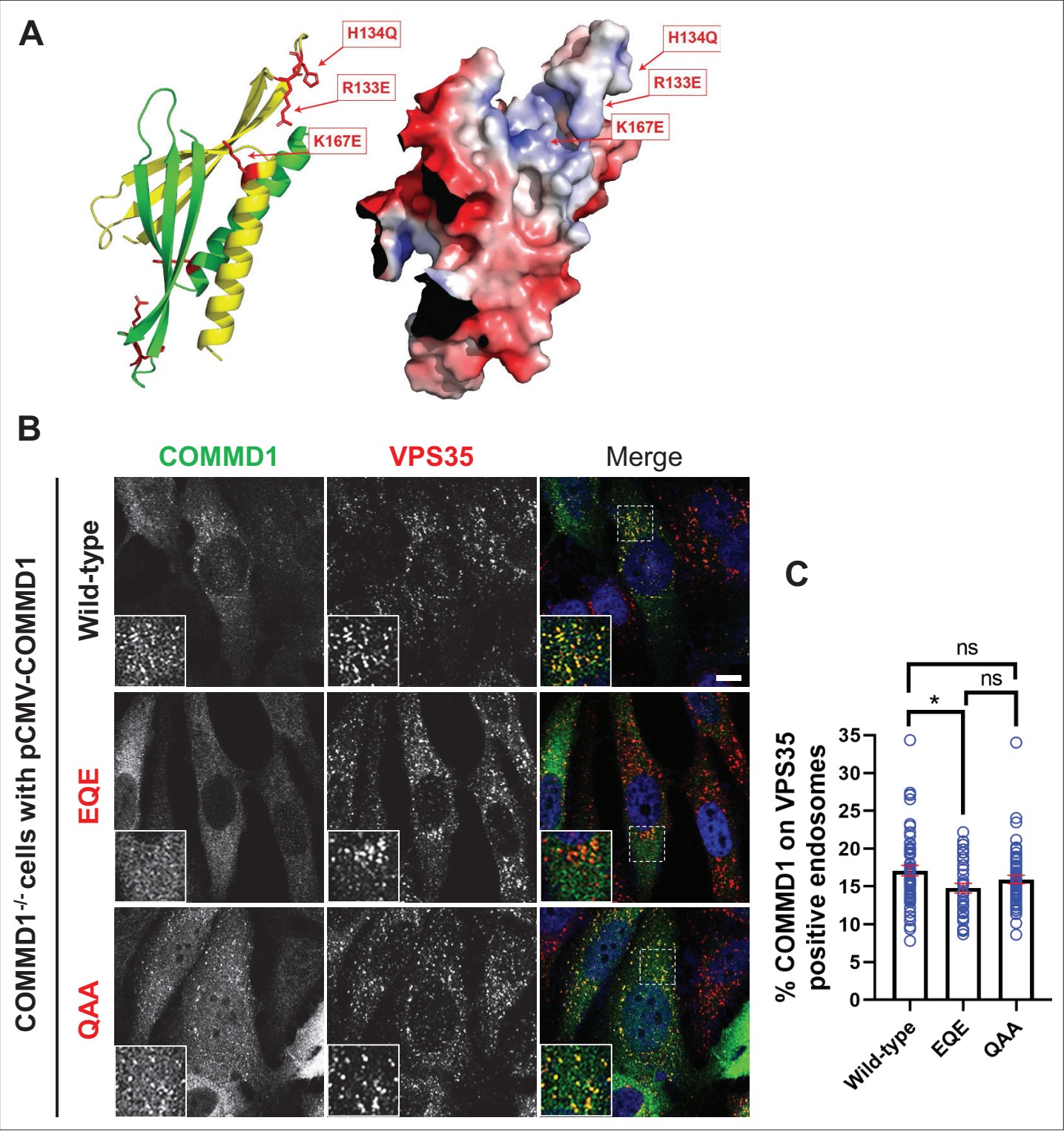

**Figure 8.** Mutation of the putative phosphoinositide binding site impairs COMMD1 localization to VPS35-positive endosomes.
 (**A**) Ribbon and space filling models of the COMMD domain of COMMD1 modeled on COMMD9 (PDB: 6BP6) (*Healy et al., 2018*). Positively charged residues within the predicted phosphorylated phosphatidylinositol (PPI) binding site are indicated. (**B–C**) COMMD1-/- HeLa cells were transiently transfected with either wild-type COMMD1 or COMMD1 mutants (EQE and QAA), then fixed, permeabilized, and co-stained with antibodies against COMMD1 and VPS35. EQE: R133E/H134Q/K167E, and QAA: R133Q/H134A/ K167A. The percent of the total COMMD1 residing on VPS35-positive endosomes was quantified. Data presented as mean ± SE. Statistical significance from three independent experiments were analyzed using one-way ANOVA and Tukey's post hoc test. *p < 0.05, and ns, not significant. Bar: 10 µm.

*Figure 8 continued on next page*

*Figure 8 continued*

The online version of this article includes the following source data and figure supplement(s) for figure 8:

**Source data 1.** Contains numerical source data for *Figure 8*.

**Figure supplement 1.** Validation of COMMD1-/- HeLa cells.

CCDC93, when compared with wild-type COMMD1. The difference between COMMD1-QAA and COMMD1-EQE was not statistically significant.

It is not surprising that mutation of a phosphoinositide binding site would also impair the formation of the CCC complex. Formation of the complex could potentially require association with membranes. Alternatively, this site may play a direct role in formation of the CCC complex. Thus, while it is possible that impaired association with PI3,5$P_2$ underlies the defect in COMMD1-EQE binding to endosomes, and accounts for a defect in recycling, it is also possible that the defect in the COMMD1-EQE mutant is due to a partial disruption of the CCC complex that is independent of its interaction with phospho-inositide lipids.

Importantly, the experiments showing that apilimod or siRNA depletion of PIKfyve impairs the SNX17 recycling pathway, and results in a loss of the CCC complex and Retriever complex from endosomes, suggests that PI3,5$P_2$ and PI5P play important roles in regulating the recruitment and/or function some CCC and possibly some Retriever subunits.

In addition to regulation of the SNX17-Retriever-CCC-WASH recycling pathway, PIKfyve could potentially play a role in β1-integrin recycling via control of endosome associated actin. A previous study revealed that PIKfyve negatively regulates cortactin, which in turn causes excessive Arp2/3 activity and hyperaccumulation of actin on endosomes (*Hong et al., 2015*). In those studies, PIKfyve was inhibited for 2 hr. However, in the shorter time frame of PIKfyve inhibition used in this study, we did not find any changes in the amount of cortactin that localizes to endosomes (*Figure 7—figure supplement 8*).

## Discussion

The best characterized roles of PIKfyve are its functions on lysosomes. Inhibition or depletion of PIKfyve results in enlarged lysosomes and late endosomes (*de Araujo et al., 2020*; *Dove et al., 2009*; *Ho et al., 2012*; *McCartney et al., 2014a*; *Shisheva, 2012*), and indeed, PIKfyve regulates multiple pathways that contribute to lysosome function. These include the regulation of multiple lysosomal ion channels (*Chen et al., 2017*; *Dong et al., 2010*; *Fine et al., 2018*; *She et al., 2018*; *She et al., 2019*; *Wang et al., 2017*), as well as roles in the fission and reformation of lysosomes and related organelles (*Bissig et al., 2017*; *Choy et al., 2018*; *Krishna et al., 2016*; *Yordanov et al., 2019*).

While the most apparent defects in cells following loss of PIKfyve activity are enlarged lysosomes, the multiple pleiotropic defects observed strongly suggest that PIKfyve plays key roles elsewhere in the cell. As an approach to gain mechanistic insight into roles for PIKfyve that are not lysosome-based, we sought mechanistic insight into roles for PIKfyve in cell migration.

The studies reported here reveal that PIKfyve has roles at endosomes and has a direct role in the regulation of the SNX17-Retriever-CCC-WASH complex. Moreover, endogenously tagged PIKfyve extensively colocalizes with SNX17, Retriever, CCC, and WASH complexes. PIKfyve has a robust 44% colocalization with VPS35, a subunit of the Retromer complex, as well as with the Retriever subunit, VPS35L. In addition, it is likely that PIKfyve specifically localizes to endosomes that are actively engaged in membrane transport. Early endosomes undergo rapid conversion to late endosomes (*Rink et al., 2005*), and this process is linked to Retromer and Retriever-based transport which occurs from the same membrane subdomains (*McNally et al., 2017*; *Singla et al., 2019*). Moreover, RAB5 and RAB7 act in concert to regulate Retromer recruitment to endosomes (*Seaman et al., 2009*). Thus, the hypothesis that a major pool of PIKfyve is localized to endosomal compartments involved in recycling fits with the observation that the best colocalization of PIKfyve is with VPS35, followed by good colocalization with EEA1 and RAB7.

Our mechanistic analysis of roles of PIKfyve with the SNX17-Retriever-CCC-WASH complexes relied primarily on acute inhibition, which is likely to reveal pathways where PIKfyve plays a direct role. In addition, these studies were aided by utilization of a hyperactive allele of PIKfyve. Importantly, activation of PIKfyve had the opposite effect of PIKfyve inhibition, which provides additional evidence

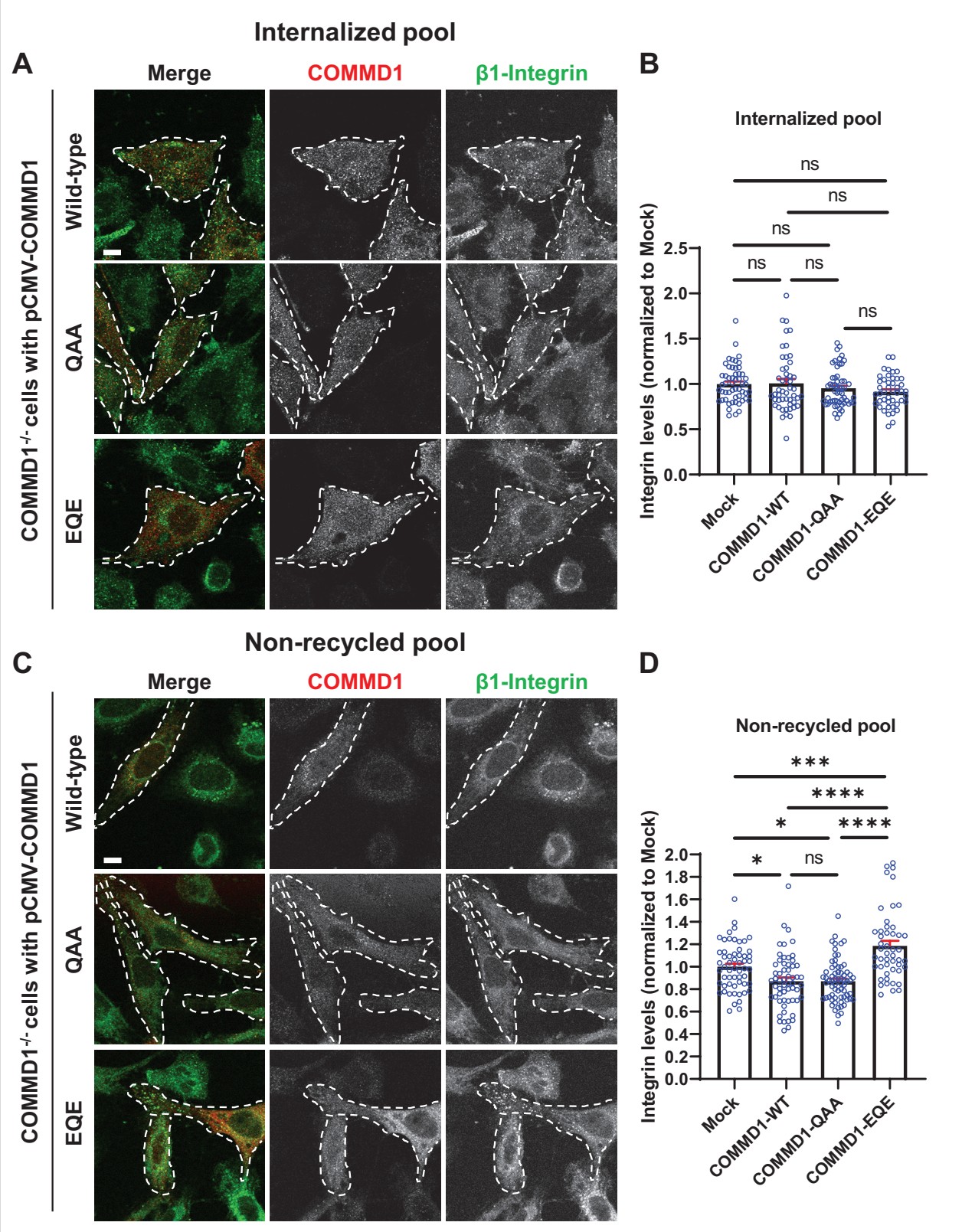

**Figure 9.** Mutation of the putative phosphoinositide binding site on COMMD1 delays recycling of β1-integrin. (A,C) COMMD1-/- HeLa cells were infected for 24 hr with lentivirus expressing either wild-type COMMD1, or the COMMD1-QAA (R133Q/H134A/K167A) or COMMD1-EQE (R133E/H134Q/K167E) mutants. Cells were then incubated with anti-β1-integrin antibodies for 1 hr in serum containing media, then acid washed and either fixed (**A and B**) or incubated again with serum containing media for 1 hr then acid washed and fixed (**C and D**). Bar: 10 μm. (B,D) Intensity of integrin was quantified

*Figure 9 continued on next page*

*Figure 9 continued*

from three independent experiments and values were normalized to the corresponding average intensity of the mock treatment cohort. Data presented as mean ± SE. Statistical significance was analyzed using one-way ANOVA and Tukey's post hoc test. *p < 0.05, ***p < 0.005, ****p < 0.001, and ns, not significant. Transfected cells highlighted with white dotted lines.

The online version of this article includes the following source data and figure supplement(s) for figure 9:

**Source data 1.** Contains numerical source data for *Figure 9*.

**Figure supplement 1.** Mutation of the putative phosphoinositide binding site on COMMD1 partially impairs the interaction with CCDC93.

**Figure supplement 1—source data 1.** Contains numerical and uncropped western blot source data for *Figure 9—figure supplement 1*.

that PI3,5P$_2$ and/or PI5P play direct roles in the SNX17-Retriever-CCC-WASH pathway, and in β1-integrin recycling.

Results reported here, together with earlier studies reveal that the SNX17-Retriever-CCC-WASH recycling pathway is ordered by changes in phosphoinositide lipids as well as a web of protein-protein interactions (*Figure 10*). VPS34, which resides on endosomes (*Christoforidis et al., 1999*), provides the PI3P (*Figure 7*, *Figure 7—figure supplement 1*) for recruitment of SNX17 (*Chandra et al., 2019*; *Jia et al., 2014*), where it can then bind its cargoes (*Böttcher et al., 2012*; *Steinberg et al., 2012*). The generation of PI3P also recruits PIKfyve, via its FYVE domain (*Stenmark et al., 2002*), which initiates the production of PI3,5P$_2$ and PI5P. Note that PI3P, or PI3,5P$_2$ and PI5P also play a role in the endosomal localization of the WASH complex (*Jia et al., 2010*) (and *Figure 7*, *Figure 7—figure supplement 4*). The endosomal association of the WASH complex also requires the Retromer complex (*Harbour et al., 2010*; *Harbour et al., 2012*; *Helfer et al., 2013*; *Jia et al., 2012*; *Zavodszky et al., 2014*). The generation of PI3,5P$_2$ and/or PI5P then facilitates the recruitment of the CCC complex (*Figure 7*). The CCC complex also binds the WASH complex via direct interaction of CCDC93 with the WASH subunit, FAM21 (*Phillips-Krawczak et al., 2015*). The Retriever complex may associate with endosomes via directly binding to PI3,5P$_2$ and/or PI5P (*Figure 7*), and/or may indirectly require PIKfyve activity to recruit and/or stabilize the CCC complex. Note that the Retriever complex interacts with and requires the CCC complex to bind to endosomes (*McNally et al., 2017*; *Phillips-Krawczak et al., 2015*; *Singla et al., 2019*). In addition, the Retriever subunit, VPS26C, interacts directly with SNX17 (*Farfán et al., 2013*; *McNally et al., 2017*).

The studies reported here also show that SNX17 and the WASH complex can bind to endosomes without the Retriever and CCC complexes. However, endosomes that contain SNX17 and the WASH complex are not sufficient for the recruitment of either the Retriever or CCC complexes. PIKfyve activity is also needed for recruitment of Retriever and CCC complexes. Furthermore, the CCC subunit CCDC22 recruits MTMR2, which is required for late steps in this recycling pathway (*Singla et al., 2019*). Importantly, we found that recruitment of MTMR2 lowers both PI3P and PI3,5P$_2$ (*Singla et al., 2019*). Thus, the SNX17-Retriever-CCC-WASH pathway may be ordered in part via an initiation step that involves PI3P, middle steps that require PI3,5P$_2$ and/or PI5P and a late step via MTMR2 that removes PI3P and PI3,5P$_2$. Finally, once the SNX17-Retriever-CCC-WASH complex assembles with cargo, the WASH complex mediates actin nucleation (*Derivery et al., 2009*; *Gomez and Billadeau, 2009*) and SNX17 recruits EHD1 (*Dhawan et al., 2020*) to enable fission of cargo containing membrane for efficient recycling.

While we favor a model where positive regulation of the SNX17-Retriever-CCC-WASH pathway occurs via direct binding of some of the subunits within these complexes to PI3P and PI3,5P$_2$ or PI5P, it is possible that these lipids are impacting the pathway in a more global way. For example, PIKfyve is important for endosome maturation (*Kim et al., 2014*; *Messenger et al., 2015*), and this could play a role in how PIKfyve regulates the SNX17-Retriever-CCC-WASH pathway.

The new role for PIKfyve in endocytic recycling reported here may partially explain a recent study which showed that treatment of cells with 100 nM apilimod for 16 hr resulted in a reduction in steady-state levels of exogenously expressed, FLAG-TGFβ-R2 at the cell surface. Note that the recycling pathway utilized by TGFβ-R2 is currently unknown (*Cinato et al., 2021*).

Previous studies have also revealed roles for PIKfyve on endosomes. The PIKfyve pathway plays a role in the formation of Stage I melanosomes, which are derived from early endosomes (*Bissig et al., 2019*). Furthermore, we and others previously showed that knock-down of the PIKfyve pathway causes a defect in recycling of CI-MPR from endosomes to the trans-Golgi network (*de*

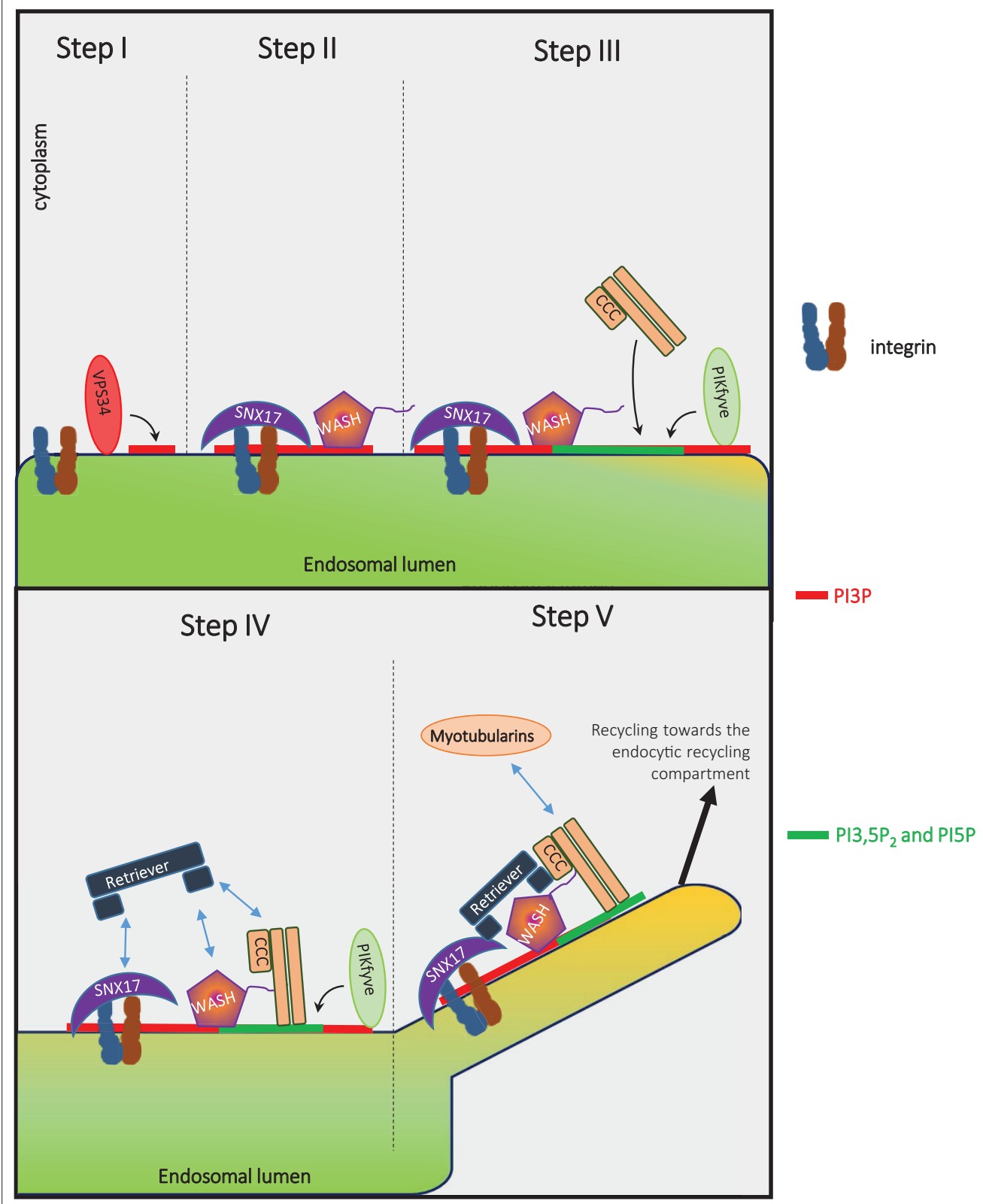

**Figure 10.** Model: VPS34 and PIKfyve regulate recycling of integrins from endosomes to the plasma membrane via promoting the ordered assembly of SNX17, WASH, Retriever, and CCC complexes. The phosphatidylinositol 3-phosphate (PI3P) on cargo-containing endosomes is generated by VPS34 (*Figure 7*, *Figure 7—figure supplement 1*). PI3P facilitates the recruitment of SNX17 (*Chandra et al., 2019*; *Jia et al., 2014*), and SNX17 binds its cargo (*Böttcher et al., 2012*; *Steinberg et al., 2012*). The generation of PI3P likely recruits PIKfyve, via its FYVE domain (*Stenmark et al., 2002*), which

*Figure 10 continued on next page*

*Figure 10 continued*

initiates the production of PI3,5P$_2$ and phosphatidylinositol 5-phosphate (PI5P). The generation of PI3P, PI3,5P$_2$, and/or PI5P plays a role in the binding of the WASH complex: WASH also requires the retromer to be recruited to endosomes (*Harbour et al., 2010*; *Harbour et al., 2012*; *Helfer et al., 2013*; *Jia et al., 2012*; *Zavodszky et al., 2014*). The generation of PI3,5P$_2$ and/or PI5P by PIKfyve then plays a role in recruitment of the CCC complex (*Figure 7*). The CCC complex also binds the WASH complex via direct interaction of CCDC93 with the WASH subunit, FAM21 (*Phillips-Krawczak et al., 2015*). The Retriever complex may also associate with endosomes by directly binding to PI3,5P$_2$ or PI5P, and/or may require PIKfyve activity to recruit the CCC complex. The Retriever complex interacts with and requires the CCC complex to bind to endosomes (*McNally et al., 2017*; *Phillips-Krawczak et al., 2015*; *Singla et al., 2019*). In addition, the Retriever subunit VPS26C interacts directly with SNX17 (*Farfán et al., 2013*; *McNally et al., 2017*). Importantly, while SNX17 and WASH are necessary, they are not sufficient for the recruitment of either Retriever or CCC (*Figure 7*, apilimod treatment). Recruitment of the Retriever and CCC complexes requires PIKfyve. Furthermore, the CCC subunit CCDC22 in turn recruits MTMR2, which is required for late steps in this recycling pathway (*Singla et al., 2019*). Importantly, recruitment of MTMR2 lowers PI3P and PI3,5P2 (*Singla et al., 2019*). Together, these studies suggest that PI3P and PI3,5P$_2$ coordinate the SNX17-Retriever-CCC-WASH pathway. PI3P is required for initiation, PI3,5P$_2$ and/or PI5P act in the middle steps, and MTMR2, which removes PI3P and PI3,5P$_2$, acts late in the pathway. Once the SNX17-Retriever-CCC-WASH complex assembles with cargo, the WASH complex mediates actin nucleation (*Derivery et al., 2009*; *Gomez and Billadeau, 2009*) and SNX17 recruits EHD1 (*Dhawan et al., 2020*) to enable fission of cargo containing membranes for efficient recycling.

*Lartigue et al., 2009*; *Rutherford et al., 2006*; *Zhang et al., 2007*), although the mechanistic basis of this defect remains unknown. Moreover, a recent study revealed that SNX11, which plays a role in the delivery of some cargoes from endosomes to lysosomes, binds to PI3,5P$_2$ (*Xu et al., 2020*). In addition, PIKfyve inhibition caused the accumulation of the tight junction proteins, claudin1 and claudin2, into endosomes and delayed the formation of epithelial permeability barrier (*Dukes et al., 2012*). Interestingly, PIKfyve inhibition did not affect the surface localization of claudin4 or the surface levels of EGFR (*Figure 6—figure supplement 2*). This suggests that PIKfyve regulates specific recycling pathways. Additionally, heterologous studies in *Xenopus laevis* oocytes suggested a connection between PIKfyve activity and RAB11 endosomes that regulate endocytic recycling (*Seebohm et al., 2012*; *Seebohm et al., 2007*). Furthermore, we previously found that PIKfyve provides acute regulation of the levels of the α-amino-3-hydroxy-5-methyl-4-isoxazolepropionic acid receptor (AMPAR) at neuronal postsynaptic sites (*McCartney et al., 2014b*; *Zhang et al., 2012*). Those studies suggested that PIKfyve is a negative regulator of AMPAR recycling, which is the opposite of PIKfyve providing positive regulation of α5-integrin, β1-integrin, and LRP1 recycling. The difference may be that while these latter proteins traffic via the SNX17-Retriever-CCC-WASH pathway, AMPAR recycles via the SNX27-Retromer pathway (*Hussain et al., 2014*). Thus, PIKfyve may differentially regulate these two pathways. Alternatively, the role of PIKfyve in recycling from endosomes to the plasma membrane in neurons may be different from the role of PIKfyve in other cell types.

In addition to regulating surface levels of β1-integrin, PIKfyve likely regulates cell migration via control of additional pathways. Previous studies suggested that PIKfyve may also regulate cell migration via the activation of the Rac1-GTPase (*Dayam et al., 2017*; *Oppelt et al., 2014*). PIKfyve regulation of Rac1 activation and integrin recycling may occur in parallel. In addition, there is extensive cross-talk between Rho GTPases and focal adhesions (*Vitali et al., 2019*), thus integrin trafficking and Rac1 activation may be linked with each other.

Apilimod, the PIKfyve inhibitor used in these studies, has recently been proposed as a drug to investigate further for the treatment of COVID-19 (*Kang et al., 2020*; *Ou et al., 2020*; *Riva et al., 2020*), and also blocks the infection of Ebola virus in cells (*Nelson et al., 2017*; *Qiu et al., 2018*). Apilimod may block SARS-CoV-2 entry via loss of positive regulation of TPC2, a downstream target of PIKfyve that resides on lysosomes (*Ou et al., 2020*). The new studies presented here suggest an additional potential mechanism and indicate that the effect of apilimod on surface levels of angiotensin-converting enzyme 2 (ACE2) and Neuropilin-1 should also be investigated. ACE2 and Neuropilin-1 levels at the plasma membrane are regulated. Importantly, each have been proposed to serve as receptors for SARS-CoV-2 (*Daly et al., 2020*; *Wrapp et al., 2020*). Moreover, a genome-wide screen to identify proteins that regulate entry of SARS-CoV2 identified subunits of the CCC, Retriever, and WASH complexes (*Zhu et al., 2021*). Together, the potential of apilimod as an antiviral drug heightens the urgency of determining the multiple cellular functions of PIKfyve, and the elucidation of pathways that are impacted by acute inhibition of PIKfyve.

# Materials and methods

**Key resources table**

| Reagent type (species) or resource | Designation | Source or reference | Identifiers | Additional information |
|---|---|---|---|---|
| antibody | Anti-HA (mouse monoclonal) | Cell signaling | 2,367 (6E2) RRID:AB_10691311 | IF (1:100) |
| antibody | Anti-HA (Rabbit monoclonal) | Cell signaling | 3,724 (C29F4) RRID:AB_1549585 | IF and WB (1:2000) |
| antibody | CCDC93 (Rabbit polyclonal) | Proteintech | 20861-1-1AP RRID:AB_10696446 | IF (1:200) and WB (1:1000) |
| antibody | COMMD1 (mouse monoclonal) | R&D systems | MAB7526 RRID:AB_2895087 | IF (1:100) |
| antibody | COMMD1 (Rabbit polyclonal) | Proteintech | 11938–1-AP RRID:AB_2083542 | IF (1:100) |
| antibody | COMMD5 (Rabbit polyclonal) | Proteintech | 10393–1-AP RRID:AB_2083555 | IF (1:100) |
| antibody | Cortactin (mouse monoclonal) | Millipore | 05–180 (4 F11) RRID:AB_309647 | IF (1:100) |
| antibody | EEA1 (mouse monoclonal) | BD biosciences | 610456 RRID:AB_397829 | IF (1:25) |
| antibody | EEA1 (Rabbit monoclonal) | Cell signaling | 3,288 (C45B10) RRID:AB_2096811 | IF (1:100) |
| antibody | FAM21 (Rabbit polyclonal) | Daniel D. Billadeau | N/A | IF (1:1000) |
| antibody | Lamp1 (mouse monoclonal) | Developmental Hybridoma | H4A3 RRID:AB_2296838 | IF (1:50) |
| antibody | Lamp1 (Rabbit polyclonal) | Abcam | ab24170 RRID:AB_775978 | IF (1:1000) |
| antibody | Lamp2a (mouse monoclonal) | BD biosciences | 555,803 (H4B4) RRID:AB_396137 | IF (1:100) |
| antibody | LRP1 (Rabbit polyclonal) | Cell signaling | 64099 RRID:AB_2799654 | WB (1:1000) |
| antibody | PIKfyve (Rabbit polyclonal) | Proteintech | 13361–1-AP RRID:AB_10638310 | WB (1:1000) |
| antibody | RAB11 (Rabbit monoclonal) | Abcam | ab128913 RRID:AB_11140633 | IF (1:200) |
| antibody | RAB4 (Rabbit polyclonal) | Abcam | ab13252 RRID:AB_2269374 | IF (1:200) |
| antibody | RAB5 (Rabbit polyclonal) | Abcam | ab18211 RRID:AB_470264 | IF (1:200) |
| antibody | RAB7 (Rabbit monoclonal) | Abcam | ab137029 RRID:AB_2629474 | IF (1:100) |
| antibody | SNX17 (Rabbit polyclonal) | Protein Atlas | HPA043867 RRID:AB_10961129 | IF (1:100) |
| antibody | Strumpellin (Rabbit polyclonal) | Bethyl Laboratories | A304-809A RRID:AB_2621004 | IF (1:100) |
| antibody | Transferrin receptor (Rabbit polyclonal) | Proteintech | 10084–2-AP RRID:AB_2240403 | WB (1:2000) |
| antibody | VPS35 (Goat polyclonal) | Abcam | ab10099 RRID:AB_296841 | IF (1:500) |
| antibody | VPS35L (Rabbit polyclonal) | Daniel D. Billadeau | N/A | IF (1:500) |

*Continued on next page*

*Continued*

| Reagent type (species) or resource | Designation | Source or reference | Identifiers | Additional information |
|---|---|---|---|---|
| antibody | α5-integrin (Rabbit polyclonal) | Cell signaling | 4705 RRID:AB_2233962 | WB(1:1000) |
| antibody | α5-integrin (mouse monoclonal) | BD Biosciences | 555615 RRID:AB_395982 | assay 5 µg/ml |
| antibody | β1-integrin (mouse monoclonal) | Millipore | MAB2000 (HB1.1) | assay 5 µg/ml |
| antibody | β1-integrin (mouse monoclonal) | Santa Cruz Biotechnology | sc-13590 (P5D2) RRID:AB_627008 | assay 5 µg/ml |
| antibody | β1-integrin (Rabbit polyclonal) | Cell signaling | 4706 RRID:AB_823544 | WB(1:1000) |
| chemical compound, drug | Apilimod | Axon Medchem | Axon1369 | 1 µM |
| chemical compound, drug | VPS34-IN1 | EMD | 532,628 | 0.1 µM, 1 µM |
| chemical compound, drug | DMEM | Thermo Fisher Scientific | 11,995 | |
| chemical compound, drug | FBS | Sigma | F4135 | |
| chemical compound, drug | penicillin/ streptomycin/ glutamate | Life Technologies | 10,378 | |
| chemical compound, drug | Blasticidin S HCl | Life Technologies | A11139 | |
| chemical compound, drug | Hygromycin B | Life Technologies | 10,687 | |
| chemical compound, drug | Doxycycline | Sigma | D9891 | |
| chemical compound, drug | Fibronectin | Sigma | F1141 | |
| chemical compound, drug | Laminin | Gibco | 23017015 | |
| commercial assay or kit | FuGENE 6 | Promega | E2691 | |
| commercial assay or kit | Lipofectamine 2000 | Thermo Fisher Scientific | 11668027 | |
| commercial assay or kit | Dharmafect 1 | Horizon Discovery Biosciences | T-2001 | |
| commercial assay or kit | Click-iT EdU Imaging Kits | Invitrogen | C10337 | |
| commercial assay or kit | LIVE/DEAD Viability/Cytotoxicity Kit | Invitrogen | L3224 | |
| commercial assay or kit | pGEM-T-Easy vector system | Promega | A1360 | |
| recombinant DNA reagent | Plasmid: pHA-CMV | Clontech | 631,604 | |
| recombinant DNA reagent | Plasmid: 6xHA-PIKfyve | Weisman Lab | See Materials and Methods. Available from LSW. | |
| recombinant DNA reagent | Plasmid: 6xHA-PIKfyve-KYA | Weisman Lab | See Materials and Methods. Available from LSW. | |
| recombinant DNA reagent | Plasmid: pSpCas9(BB)–2A-GFP | Addgene | 48,138 | |

*Continued on next page*

*Continued*

| Reagent type (species) or resource | Designation | Source or reference | Identifiers | Additional information |
|---|---|---|---|---|
| recombinant DNA reagent | pLenti-Lox-EV-3XHA-COMMD1 | Weisman Lab | | See Materials and Methods. Available from LSW. |
| recombinant DNA reagent | pLenti-Lox-EV-3XHA-COMMD1-QAA (R133Q/ H134A/ K167A) | Weisman Lab | | See Materials and Methods Available from LSW. |
| recombinant DNA reagent | pLenti-Lox-EV-3XHA-COMMD1-EQE (R133E/ H134Q/ K167E) | Weisman Lab | | See Materials and Methods. Available from LSW. |
| cell line (*Homo-sapiens*) | Human: HeLa | Richard Klausner Lab | STR validated RRID:CVCL_0030 | |
| cell line (*Homo-sapiens*) | Human: HeLa | Burstein Lab | STR validated RRID:CVCL_0030 | |
| cell line (*Homo-sapiens*) | Human: HeLa COMMD1-/- cells | Burstein Lab | N/A | |
| cell line (*Homo-sapiens*) | Human: HEK293 | ATCC | CRL-1573 | |
| cell line (*Homo-sapiens*) | Human: HUH7 | David Ginsburg Lab (University of Michigan-Ann Arbor) | N/A | |
| cell line (*Mus-musculus*) | Mouse: Primary MEF (Wild Type) | Weisman Lab *Zolov et al., 2012* | | |
| cell line (*Mus-musculus*) | (PIKfyve$^{\beta\text{-geo}/\beta\text{-geo}}$) | Weisman Lab *Zolov et al., 2012* | | |
| cell line (*Mus-musculus*) | Mouse: Primary neonatal cardiac fibroblasts | Alan Smrcka (University of Michigan-Ann Arbor) | N/A | |
| cell line (*Homo-sapiens*) | Human: Flp-In T-Rex 293 cells | Life Technologies | R780-07 | |
| cell line (*Homo-sapiens*) | Human: Flp-In T-Rex 293-3XFLAG | Weisman Lab *McCartney et al., 2014b* | | |
| cell line (*Homo-sapiens*) | Human: Flp-In T-Rex 293-3XFLAG-Citrine-PIKfyve | Weisman Lab *McCartney et al., 2014b* | | |
| cell line (*Homo-sapiens*) | Human: Flp-In T-Rex 293-3XFLAG-Citrine-PIKfyve-KYA | Weisman Lab *McCartney et al., 2014b* | | |
| cell line (*Homo-sapiens*) | Human: 3xHA-PIKfyve knock-in HEK293 | Weisman Lab | This manuscript. Available from LSW | |
| software, algorithm | Adobe Photoshop CS6 | Adobe Studios | N/A | |
| software, algorithm | Adobe Illustrator CS6 | Adobe Studios | N/A | |
| software, algorithm | Excel | Microsoft | N/A | |
| software, algorithm | Prism 8.1.2 | GraphPad | https://www.graphpad.com | |
| software, algorithm | ImageJ/Fiji | *Schindelin et al., 2012* | https://imagej.net/Fiji | |
| software, algorithm | CellProfiler | *Lamprecht et al., 2007*; *Schindelin et al., 2012* | https://cellprofiler.org | |
| software, algorithm | Image Lab 6.0.0 | Bio-Rad | https://www.bio-rad.com/ | |

## Cell culture, transfection, and plasmids

Cells were tested for mycoplasma using LookOut Mycoplasma PCR Detection Kit (Sigma Aldrich, Cat. MP0035). HeLa, HEK293, HUH7 cell lines were grown in DMEM supplemented with 10% FBS, penicillin/streptomycin/glutamate (PSG). Primary MEF cells were cultured in DMEM containing 15% FBS and PSG. Flp-In T-Rex 293 cells that inducibly express a cDNA encoding 3xFLAG alone, 3xFLAG-Citrine-PIKfyve, or 3xFLAG-Citrine-PIKfyve-KYA have been previously described (*McCartney et al., 2014b*). These cell lines were grown in DMEM supplemented with 10% FBS, PSG, 15 µg/ml Blasticidin S HCl, and 0.4 mg/ml Hygromycin B, and cells were induced with 100 ng/ml doxycycline for 12 hr. For transient transfection of PIKfyve, FuGENE 6 was used following manufacturer's instructions and transfection time was 16–18 hr. All cells were cultured in 5% $CO_2$ at 37°C. For immunofluorescence studies,

HEK293 and 3xHA-PIKfyve knock-in HEK293 cells were grown in fibronectin and laminin-coated coverslips. siGENOME siRNAs D-005058-10-0010 and D-005058-12-0010 targeting the 5'UTR region of PIKfyve was used to deplete PIKfyve. Cells were transfected with 5 µM of both siRNAs for 3 days. 6XHA-PIKfyve and 6xHA-PIKfyve-KYA were generated by cloning PIKfyve into pHA-CMV and adding 5XHA tag using Gibson assembly as described previously (*Cinato et al., 2021*). HeLa COMMD1 knockout cells were generated as previously described (*Singla et al., 2019*) using CRISPR/Cas9-mediated gene deletion and the two COMMD1 target guides: 5'-ACGGAGCCAGCTATATCCAG-3' and 5'-GCGCATTCAGCAGCCCGCTC-3'.

3XHA-COMMD1, 3XHA-COMMD1-QAA, 3XHA-COMMD1-EQE, and 3XHA-COMMD1-EDEQE constructs were generated by cloning cDNA of 3XHA-COMMD1 and its mutants to pLenti-Lox-EV vector using Gibson assembly. COMMD1 knockout cells were transiently transfected with these COMMD1 constructs using Lipofectamine 2000 following manufacturer's instructions for 24 hr.

## Generation of cells with endogenous expression of 3xHA-PIKfyve

HEK293 cells were modified by CRISPR-Cas9 genome editing to add a 3xHA tag to PIKfyve at the N-terminus. Donor DNA spanning 315 base pairs on the left homologous region and 353 base pairs on the right homologous region was generated by overlap extension PCR and then cloned into the pGEM-T-Easy vector system. Guide RNA (TGATAAGACGTCCCCAACAC) for PIKfyve was cloned into pX458, a pSpCas9-2A-EGFP vector. Lipofectamine 2000 was used for transfection of cells with pX458 expressing Cas9 along with a gRNA donor vector. Three days after transfection, GFP-positive single cells were sorted using flow cytometry into a 96-well plate containing conditioned media. 3xHA-PIKfyve knock-in HEK293 cells were validated by PCR, the PCR product was sequence verified, and the cell lysate was verified by western blot.

## Wound healing and cell spreading assays

Cells were grown on coverslips or fibronectin-coated plastic dishes to full confluency and were wounded using a pipet tip, then incubated in DMEM with 10% FBS for the indicated treatments and time points. Quantification of wound healing was performed on five random fields for each condition and wound area closure was quantified from three independent experiments. For cell spreading assays, cells were trypsinized, seeded onto fibronectin-coated plastic dishes, and allowed to attach to dishes in DMEM with 10% FBS for the indicated times.

## Cell proliferation and cell viability assay

Cell proliferation assays were performed with Click-iT EdU Imaging Kits. HeLa cells grown on coverslips with treatments as indicated were incubated with 10 µM EdU (5-ethynyl-2'-deoxyuridine) in DMEM with 10% FBS for 30 min. Cells were fixed, permeabilized, and incubated with Click-iT reaction cocktail to detect the incorporated EdU according to manufacturer's instructions. Cells were mounted and analyzed. Cells which incorporated Edu were identified as proliferating.

Cell viability assays were performed using LIVE/DEAD Viability/Cytotoxicity Kit, for mammalian cells. HeLa cells grown on coverslips with treatments as indicated were incubated in DMEM with 10% FBS containing 2 µM calcein AM and 4 µM ethidium homodimer-1 (EthD-1) for 30 min. Cells were mounted with 10 µl PBS and analyzed by fluorescence microscopy.

## Surface biotinylation

Cells were incubated with DMSO or 1 µM apilimod for 1 hr at 37°C, then transferred to 4°C for 15 min. Cells were washed with ice-cold wash buffer (PBS containing 2.5 mM MgCl$_2$ and 1 mM CaCl$_2$) and incubated with ice-cold 0.5 µg/ml NHS-SS-Biotin (Pierce) for 20 min. Biotinylation was quenched by incubating the cells with ice-cold 100 mM glycine for 10 min. Cells were then pelleted, lysed in RIPA buffer (Pierce) containing protease and phosphatase inhibitors. Three mg of protein lysate was incubated overnight with 100 µl of streptavidin bead slurry. Western blot analysis was performed on 20% of the total of each immunoprecipitate and 50 µg of each lysate.

## Labeling of surface-exposed integrin

HeLa cells after indicated treatments were incubated with ice-cold serum-free media containing 0.5% BSA and 5 µg/ml mouse anti-β1-integrin antibody (MAB2000) or mouse anti-α5-integrin (555615) for

1 hr at 4°C. Cells were fixed in ice-cold 4% paraformaldehyde for 30 min at 4°C. Cells were permeabilized and immunostained with Alexa-Fluor 488-conjugated donkey anti-mouse secondary antibodies. To determine the effect of PIKfyve mutants on the surface levels of integrin (*Figure 1H*), cells were fed with complete media for 2 hr prior to labeling surface integrin. Cells were then incubated with 5 µg/ml mouse anti-β1-integrin antibody for 1 hr at 4°C. Cells were fixed at 4°C for 30 min and permeabilized. Cells were immunostained with rabbit anti-HA antibodies followed by Alexa-Fluor 488-conjugated donkey anti-mouse and Alexa-Fluor 488-conjugated goat anti-rabbit secondary antibodies.

## Integrin trafficking experiments

To determine the dynamic regulation of integrin levels at the cell surface, cells were incubated with serum-free media containing 0.5% BSA and 5 µg/ml mouse anti-β1-integrin antibody (MAB2000) for 1 hr at 4°C. Cells were then incubated with DMSO or 1 µM apilimod in DMEM with 10% FBS at 37°C for the indicated times. To measure the surface levels, cells were fixed with 4% paraformaldehyde after treatment. To determine the internalized and unrecycled pool, cells were fixed after a brief acid wash of 0.5% acetic acid and 0.5 M NaCl for 1 min, to remove surface antibodies. Fixed cells were immunostained with secondary antibodies and analyzed by confocal microscope.

To measure β1-integrin recycling, cells were incubated with 5 µg/ml mouse anti-integrin antibody (P5D2) in DMEM with 10% FBS for 1 hr at 37°C. Cells were acid washed with PBS, pH 3.0 for 1 min. At this stage, cells were either fixed with 4% paraformaldehyde to determine the internalize pool of β1-integrin or the following treatments were performed. Cells were then either untreated or incubated with DMSO or 1 µM apilimod in DMEM with 10% FBS at 37°C for the indicated times. To determine the non-recycled pool, cells were fixed after a brief acid wash with PBS, pH 3.0 for 1 min. Fixed cells were incubated with the indicated antibodies and analyzed. Cells were co-labeled with Texas Red-WGA to mark the cell border. To determine the localization of non-recycled integrin, cells were co-incubated with antibodies to the endocytic markers: EEA1 for early endosome, LAMP1 for late endosome and lysosome, RAB4 for fast recycling endosomes ,and RAB11 for slow recycling endosomes.

To measure the effect of COMMD1 mutants on β1-integrin trafficking, COMMD1-/- cells were first infected with lentivirus expressing the COMMD1 mutants or wild-type COMMD1 for 24 hr. Cells were incubated with 5 µg/ml mouse anti-integrin antibody (P5D2) in DMEM with 10% FBS for 1 hr at 37°C. Cells were acid washed. Cells were either fixed or fixed after incubation with DMEM with 10% FBS for 1 hr at 37°C, then followed with a brief acid wash.

## Immunofluorescence and image acquisition

For PIKfyve colocalization studies, cells were serum starved for 2 hr and fed with DMEM with 10% FBS for 30 min prior to fixation; other cells were fixed directly after the indicated treatments. Cells were fixed with 4% paraformaldehyde for 10 min at room temperature unless specified. Fixed cells were permeabilized with PBS containing 0.5% BSA and 0.2% saponin. Permeabilized cells were incubated with the indicated antibodies and analyzed. Images were acquired with an Olympus FV1000, LEICA SP5, or LEICA SP8 confocal microscope under an oil immersion 60× or 63× objective, respectively. Immunoblots were imaged using a ChemiDoc Imaging System (Bio-Rad).

## Image analysis

Images were analyzed using Fiji (ImageJ; NIH) or CellProfiler. To determine the intensity of integrin during trafficking and surface labeling experiments, cells were segmented using the ImageJ crop function and integrated density was measured using the analyze function. The changes in the surface levels of integrin were inferred from the percentage of integrin intensity within 0.8 µm from the plasma membrane; the enlarge function was employed to mark this area. To measure the intensity of proteins on VPS35-positive endosomes, for each field of cells, a mask was created in ImageJ using VPS35 fluorescence and then overlaid onto the fluorescence channel of the protein of interest.

To determine the colocalization between integrin and endocytic markers, images were segmented, and colocalization was measured using CellProfiler. To determine the colocalization of PIKfyve with other proteins or markers, images were cropped and the colocalization was measured using the Jacop plugin-in, ImageJ. Immunoblots were analyzed using Image Lab Software.

## Flow cytometry

To measure the surface levels of β1-integrin, HeLa cells treated with DMSO or 1 μM apilimod for 1 hr were trypsinized and strained through a 70 μm strainer to remove aggregated cells. Then, cells were washed once with cold PBS and incubated with cold serum-free media containing 5 μg/ml mouse anti-β1-integrin antibody (MAB2000) for 30 min on ice. Cells were washed once again with cold PBS and incubated cold serum-free media containing Alexa-Fluor 568-conjugated donkey anti-mouse secondary antibody for 30 min on ice. After washing once again with cold PBS, cells were fixed with 4% formaldehyde at room temperature for 10 min, pelleted and resuspended in PBS; 10,000 cells were analyzed per experiment.

## Cell permeabilization

Cell permeabilization was tested as an approach to determine the fraction of SNX17-related proteins that remain associated with membranes following treatment with inhibitors. HeLa cells after the indicated treatments for 2 hr were either unpermeabilized or permeabilized with 1× PBS containing 100 μg/ml digitonin for 2 min at room temperature. Cells were then washed with 1× PBS and cells were extracted in 2× Laemmli buffer containing β-mercaptoethanol. Cell lysates were immunoblotted.

## Immunoprecipitation

HeLa cells were infected with lentivirus expressing 3xHA-COMMD1, or 3xHA-COMMD1-QAA (R133Q/H134A/K167A) or 3xHA-COMMD1-EQE (R133E/H134Q/K167E) mutants for 48 hr. Cells were lysed with Pierce IP lysis buffer (ThermoFisher, 87787) supplemented with protease inhibitors (cOmplete Mini, EDTA-free, 11836170001, Roche) and phosphatase inhibitor (PhosSTOP, 04906845001 Roche). Cell lysates were incubated with Goat anti-HA antibody agarose immobilized bead (Bethyl lab, S190-138) overnight at 4°C. Precipitates were washed three times with the lysis buffer and resuspended in Laemmli sample buffer. Cell lysates and immunoprecipitant were immunoblotted with CCDC93 and HA antibodies.

## Statistical analysis

All experiments were performed at least three times. Statistical analyses are described in the figure legends. Statistical analyses were performed in GraphPad Prism 8.1.2.

## Acknowledgements

We thank Bagyasree Jambunathan and Madison Tluczek for their assistance in image analysis. We thank Aaron D Cohen, as well as members of the Weisman lab for helpful discussions. We thank Loren Brown and Alan Smrcka (University of Michigan-Ann Arbor) for generously providing primary neonatal cardiac fibroblasts. We thank David Ginsburg Lab (University of Michigan-Ann Arbor) for providing HUH7 cells. This work was supported by R01-NS099340 and R01 NS064015 to LSW, R01-DK107733 to EB and DDB, and by the University of Michigan Protein Folding Diseases Fast Forward Initiative. SSPG was supported in part by a postdoctoral fellowship from the American Heart Association, 14POST20480137. GL was supported in part by a postdoctoral fellowship from the American Heart Association, 19POST34450253.

## Additional information

### Funding

| Funder | Grant reference number | Author |
| --- | --- | --- |
| National Institute of Neurological Disorders and Stroke | Research Project Grant R01 NS064015 | Lois S Weisman |
| National Institute of Neurological Disorders and Stroke | Research Project Grant R01-NS099340 | Lois S Weisman |

| Funder | Grant reference number | Author |
|---|---|---|
| National Institute of Diabetes and Digestive and Kidney Diseases | Research Project Grant R01-DK107733 | Ezra Burstein Daniel D Billadeau |
| American Heart Association | Postdoctoral Fellowship 14POST20480137 | Sai Srinivas Panapakkam Giridharan |
| American Heart Association | Postdoctoral Fellowship 19POST34450253 | Guangming Luo |
| University of Michigan Protein Folding Diseases Fast Forward Initiative | Pilot grant | Lois S Weisman |

The funders had no role in study design, data collection and interpretation, or the decision to submit the work for publication.

## Author contributions

Sai Srinivas Panapakkam Giridharan, Guangming Luo, Conceptualization, Funding acquisition, Investigation, Methodology, Writing – review and editing; Pilar Rivero-Rios, Investigation, Writing – review and editing; Noah Steinfeld, Formal analysis, Writing – review and editing; Helene Tronchere, Amika Singla, Ezra Burstein, Daniel D Billadeau, Michael A Sutton, Investigation; Lois S Weisman, Conceptualization, Funding acquisition, Investigation, Project administration, Supervision, Writing – review and editing

## Author ORCIDs

Sai Srinivas Panapakkam Giridharan http://orcid.org/0000-0002-6214-0397
Guangming Luo http://orcid.org/0000-0003-1234-3388
Ezra Burstein http://orcid.org/0000-0003-4341-6367
Michael A Sutton http://orcid.org/0000-0003-1593-727X
Lois S Weisman http://orcid.org/0000-0001-7740-9785

## Ethics

This study was performed in strict accordance with the recommendations in the Guide for the Care and Use of Laboratory Animals of the National Institutes of Health. All the animals were handled according to approved institutional animal care and use committee (IACUC) protocols (#A3114-01) of the University of Michigan. The protocol was approved by the Committee on the Ethics of Animal Experiments of the University of Michigan (Approval # PRO00010100).

## Decision letter and Author response

Decision letter https://doi.org/10.7554/eLife.69709.sa1
Author response https://doi.org/10.7554/eLife.69709.sa2

## Additional files

### Supplementary files
• Transparent reporting form

### Data availability
All data generated or analyzed during this study are included in the manuscript and supporting files.

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
