## [Editor Report]

The authors investigate the role of the PI3P 5-kinase protein (PIKfyve) in endosome to cell surface recycling. They report that PIKfyve function is necessary for cell migration and endosomal recycling of integrin proteins via the SNX17-Retriever pathway. The manuscript will be of broad interest to researchers studying the regulation of integrin trafficking during cell migration, the organization and function of the endosomal network, and the increasingly important area of endosomal sorting de-regulation in a wide array of human diseases.

---

## [Decision Letter]

**Decision letter after peer review:**

Thank you for submitting your article "Lipid kinases VPS34 and PIKfyve coordinate a phosphoinositide cascade to regulate Retriever-mediated recycling on endosomes" for consideration by *eLife*. Your article has been reviewed by 3 peer reviewers, and the evaluation has been overseen by Suzanne Pfeffer as the Senior and Reviewing Editor. The reviewers have opted to remain anonymous.

Essential revisions:

You will see from the included comments that the reviewers are positive but request specific control experiments to enhance the impact of your story. The reviewers request specificity controls to add information related to pathway specificity, and improved cell surface analysis, localisation experiments, and the need to consider other interpretation of the relatively small phenotypes to make the model more realistic. You will also see experiments proposed to monitor active versus inactive integrins, or MTOC reorganization that are not essential. I include the full reports as the reviewers have offered detailed comments to improve the current story.

*Reviewer #1 (Recommendations for the authors):*

Overall, this study elucidates important additional steps in retriever recruitment and function but the authors should address the individual points listed below.

Figure 1: Why do the authors only use chemical inhibition of PIKfyve for their migration analysis? A simple knockdown and or knockout of PIKfyve would be the better choice, to be validated by chemical inhibition. Or do the authors think that knockdown/knockout could lead to pleiotropic effects masking the effect on recycling?

Figure 1E-H: From my personal experience with Integrin β 1 and HeLa cells I know that integrin levels are highly variable between individual Hela cells on the same microscopy specimen. Also, I don't really see any difference between the HeLa cells on the left and the PIKfyve transfected cells in the middle panel (Figure 1G). The panel on the right contains several cells which do express PIKfyve-KYA but do not display higher integrin levels. The authors need a better assay and I suggest to perform FACS analysis of surface Integrin β 1 levels. Alternatively, surface biotinylation and western blot analysis of integrin β 1 could be used. FACS is probably the best method and has been extensively used before.

Figure 7: The authors show via microscopy that Apilimod reduces the endosomal localization of retriever components. The differences appear relatively minor and since this is a key finding, it would be good if the authors could confirm this with a simple biochemical method such as western blotting for cytosol and crude membrane fractions. Again, knockdown or knockout of PIKfyve would be good to confirm these inhibitor based results.

*Reviewer #2 (Recommendations for the authors):*

In the wound healing experiments, do the authors know whether PIKfyve inhibition affects the re-polarisation of the cells at the leading edge as defined by movement of the MTOC?

The imaging based analysis of cell surface B1 integrin is consistent with PIKfyve being required for B1 integrin surface expression. These data are derived by changes in the surface levels of integrin inferred from the percentage of integrin intensity with 0.8 microns from the plasma membrane. To support this important conclusion the authors should move the biochemical data shown in Figure 6A and B to Figure 3. In addition, they should supplement these surface restricted biotinylation experiments, through inclusion of a negative control, that is a cell surface cargo whose level does not decrease upon PIKfyve inhibition (this could, for example, be a retromer pathway specific cargo protein). In this way, the authors can address whether PIKfyve is causing a global perturbation of endosomal recycling (that is, its effects are not restricted to solely the SNX17-retriever-CCC-WASH pathway) or is it having a selective effect on this pathway.

While the authors quantify that the steady-state level of B1 integrin is not reduced upon 30 minutes of PIKfyve inhibition (Figure S4), it would be informative to extend this timeline and include experiments with cycloheximide to define if integrin turnover is enhanced. This may provide insight into whether PIKfyve activity is required for integrin retrieval from lysosomal degradation or for post-retrieval transport from the endosomal compartment.

Endosomal sorting of active versus inactive forms of integrin heterodimers is an important feature of their role in cell migration. Antibodies specific for active and inactive conformation are available, and these could be incorporated into the study to provide a view of whether PIKfyve is required for the recycling of integrins independently of their activity status or is more selective for sorting of active integrins (the SNX17-retriever-CCC-WASH pathway is generally considered to support recycling of active and inactive integrin).

The quantification of endosomal association of the SNX17-retriever-CCC-WASH pathway components relies on a normalized colocalisation analysis with VPS35. A key assumption is that the endosomal association of VPS35 is not affected in anyway by the drug treatments (within the context of the rather small affects being quantified). After all, a sub-population of VPS35 is know to be associated to endosomes through binding to the PI3P-binding protein SNX3, with the remaining VPS35 population being associated through RAB7-GTP – a GTPase that itself is controlled through endosomal phosphoinositides. Teasing apart the precise roles of PI3P and PI3,5P2 in the context of endosomal association is therefore extremely difficult especially when the perturbation in localization are very small. It would perhaps be more appropriate to place less emphasis on the role of these phosphoinositides in endosome recruitment (with the exception of SNX17 and COMMD's where the evidence is perhaps more clear), by discussing the possibility that the data reflect a role for the phosphoinositide switch in aiding the timing of coupling and communication within the pathway (e.g. could be through conformational changes induced by phosphoinositide binding or altered protein orientation on the membrane).

The generation of the COMMD1-EDEQE mutant is predicted not to change its overall structure. The authors should validate that when expressed in cells this mutant can retain association to CCDC22 and CCDC93. In addition, what happens to the localization of CCDC22 and/or CCDC93 when PIKfyve is inhibited? Robust and clear loss of endosomal localization of these proteins would be needed to support the statement (line 478) 'generation of PI3,5P2 and/or PI5P then recruits the CCC complex'.

The final model needs to better reflect the complexities of endosomal association of these sorting components. The WASH complex, for example, is associated to endosomes through retromer-dependent and retromer-independent mechanisms, and while the respective level of these mechanisms may vary between cell types it is incorrect to show that WASH association is solely reliant on PI3P, PI3,5P2 and/or PI5P.

*Reviewer #3 (Recommendations for the authors):*

Suggestions for additional data/experiments:

A. There is a strong focus on the recycling of integrins and the authors do show effects on the cell surface levels of integrins and the endosome to cell surface recycling of integrins (and to a lesser degree, LRP1). I think it would be helpful however if the authors could show something that doesn't change when PIKfyve is inhibited. For example, do cell surface levels of the transferrin receptor, Glut1 or EGFR change in any way when PIKfyve in inhibited? To me, this is a necessary control that will confirm that PIKfyve has a function in sorting/transport of specific ligands from endosomes to the cell surface.

B. Similarly, when the authors have investigated effects of the expression of the VPS34-IN1 (Figure S7) they show that there are effects on every protein they investigate. Is there an endosomal protein that is not affected by the VPS34-IN1? I would also urge the authors to employ a biochemical assay to examine membrane association as relying purely on microscopy can give somewhat limited data. For example, an assay that separates membranes from cytosol by centrifugation would be useful to corroborate/support the observations made through microscopy. Certainly, some of the effects they observe when PIKfyve is inhibited are quite marginal and not necessarily consistent with PIKfyve (and PI3,5P2) playing such a key role.

Suggestions for changes to the text:

A. The authors highlight the role for PI3P in mediating the recruitment of Fam21 to endosomes but there is a more substantial body of data that show that binding of Fam21 to the retromer protein VPS35 mediates recruitment of the WASH complex to endosomes (see PMIDs: 20923837; 22070227; 23331060; 22513087; 24819384). The effects of PI3 kinase inhibition on the endosomal pool of Fam21 appear fairly marginal which might indicate that binding PI3P helps stabilise Fam21 membrane association, but it cannot be concluded that Fam21 recruitment requires PI3P. In other experiments that authors make similar claims regarding the importance of PI3P or PI3,5P2 when the effects observed as rather marginal. Toning down the text to reflect the reality of the observed effect would be beneficial.

B. Similarly, when the authors describe the role of retriever in endosome to cell surface recycling, they state that half of the cell surface proteins are trafficked by this route but what do they mean by 'half'. Are half of the many species of cell surface proteins affected by loss of retriever function or is there literally half the number of cell surface protein present at the cell surface after loss of retriever?

C. The authors should recognise (and cite) literature showing that Snx27 may also mediate endosome to cell surface recycling of integrins (see PMIDs: 27909246; 23563491). Indeed, levels of Retriever component expression (except VPS29) are often much lower than levels of retromer components suggesting that retriever may sort fewer membrane proteins (see PMID: 27278775) and that therefore, some membrane proteins could be sorted by both pathways.

[Editors' note: further revisions were suggested prior to acceptance, as described below.]

Thank you for submitting your article "Lipid kinases VPS34 and PIKfyve coordinate a phosphoinositide cascade to regulate Retriever-mediated recycling on endosomes" for consideration by *eLife*. Your article has been reviewed by 2 peer reviewers, and the evaluation has been overseen by Suzanne Pfeffer as the Senior and Reviewing Editor. The reviewers have opted to remain anonymous.

Reviewer 1 wrote, "I was disappointed by the lack of effort to biochemically verify the observed changes in membrane localization of SNX17 and retriever components. The authors state that only 15% of SNX17 and retriever components are membrane bound, which is plausible. Yet, the 15% left on membranes after digitonin permeabilization result in a good western blot signal in the panel that they provide. Please quantify the blot and add the data.

*Reviewer #1 (Recommendations for the authors):*

The authors have performed substantial new experimentation to address my concerns regarding integrin surface localization. They have also confirmed their inhibitor based results with genetic perturbation of PIKfyve, which strengthens the study.

However, I was disappointed by the lack of effort to biochemically verify the observed changes in membrane localization of SNX17 and retriever components. The authors state that only 15% of SNX17 and retriever components are membrane bound, which is plausible. Yet, the 15% left on membranes after digitonin permeabilization result in a good western blot signal in the panel that they provide. Please quantify changes in this membrane bound fraction upon Apilimod treatment. It is a key result in their study and two reviewers suggested (even "urged", see reviewer 3) that the microscopy should be confirmed by a biochemical method. The method seems to work and there are changes upon Apilimod treatment. It is a simple assay, the signal is robust enough to be quantified and the microscopy data alone is not convincing enough.

I think that the authors should repeat this fractionation a few times to quantify the changes in membrane bound SNX17 and retriever components. If this confirms their microscopy, their model is likely correct and the study publishable. If it does not confirm the microscopy, the model is probably wrong.

*Reviewer #3 (Recommendations for the authors):*

The authors have made a fair effort to address the concerns I raised previously. Whilst some issues remain unresolved, I recognize that this is likely to be due to technical challenges rather than a lack of willingness to make the necessary revisions.

Overall, I think this study has merit and will be of interest to the field. I'm not sure it is fundamentally 'game-changing' but that should not be a barrier to publication.

---

## [Author Response]

Reviewer #1 (Recommendations for the authors):Overall, this study elucidates important additional steps in retriever recruitment and function but the authors should address the individual points listed below.Figure1: Why do the authors only use chemical inhibition of PIKfyve for their migration analysis? A simple knockdown and or knockout of PIKfyve would be the better choice, to be validated by chemical inhibition. Or do the authors think that knockdown/knockout could lead to pleiotropic effects masking the effect on recycling?

PIKfyve has multiple roles and in multiple cellular location. Moreover, in mice a whole body knockout results in early embryonic lethality (PMID: 23322734, PMID: 21349843). Thus, it is likely that knockdown/knockout of PIKfyve would lead to pleiotropic effects. For this reason, we prefer short-term chemical inhibition. However, we agree that orthogonal approaches are also useful. As stated above, we now include siRNA knock-down of PIKfyve in studies of the impact on surface levels of integrins (Figures S3) and localization of COMMD1 to endosomes (Figure 7, Figure S8). In addition, we used primary fibroblasts from the hypomorphic PIKfyve b-geo/b-geo mice, a gene-trap mice that expresses only 10% of WT PIKfyve and has half the levels of PI3,5P2 and PI5P, and found that primary fibroblasts from these mice have defects in cell migration (Figure S2C-D). We also used a Flip-In cell line to over-express hyperactive PIKfyve-KYA and found that this promotes cell migration (Figure 1C-D, Figure S2G-H)

Figure 1E-H: From my personal experience with Integrin β 1 and HeLa cells I know that integrin levels are highly variable between individual Hela cells on the same microscopy specimen.

Yes, we agree that in HeLa cells integrin levels are variable. We now also include analyses of changes in the surface levels of integrin by flow cytometry, where 10,000 cells were analyzed per experiment. Similar to immunofluorescence-based analysis, treatment of apilimod lead to a decrease in the surface levels of b1-integrin by 18% (new Figure 1G).

Also, I don't really see any difference between the HeLa cells on the left and the PIKfyve transfected cells in the middle panel (Figure 1G). The panel on the right contains several cells which do express PIKfyve-KYA but do not display higher integrin levels. The authors need a better assay and I suggest to perform FACS analysis of surface Integrin β 1 levels. Alternatively, surface biotinylation and western blot analysis of integrin β 1 could be used. FACS is probably the best method and has been extensively used before.

When we chose random fields of cells for the original figure, we did not pay attention to whether the fields chosen were near the mean average for each population. We now chose representative examples of cells with intensities of integrin that are at the mean average for each population (new Figure 1I).

Figure 7: The authors show via microscopy that Apilimod reduces the endosomal localization of retriever components. The differences appear relatively minor and since this is a key finding, it would be good if the authors could confirm this with a simple biochemical method such as western blotting for cytosol and crude membrane fractions. Again, knockdown or knockout of PIKfyve would be good to confirm these inhibitor based results.

We agree that an orthogonal biochemical method to measure the changes in endosomal localization of the proteins would be ideal. However, we found that except for VPS35L, only 10-15% of SNX17, as well as each Retriever, and WASH subunit tested is associated with membranes (Figure 7—figure supplement 6). Thus, we did not pursue biochemical approaches further.

Given that only a small population is on membranes, using immunofluorescence to focus on changes at a specific compartment, for example, VPS35 containing membranes, provides a more sensitive assay.

Reviewer #2 (Recommendations for the authors):In the wound healing experiments, do the authors know whether PIKfyve inhibition affects the re-polarisation of the cells at the leading edge as defined by movement of the MTOC?

This question was tested in a previous study (PMID: 24840251), which showed that PIKfyve inhibition resulted in a depolarization of cells at the leading edge as defined by orientation of the Golgi. We now include this information with the appropriate citation in the current manuscript.

The imaging based analysis of cell surface B1 integrin is consistent with PIKfyve being required for B1 integrin surface expression. These data are derived by changes in the surface levels of integrin inferred from the percentage of integrin intensity with 0.8 microns from the plasma membrane. To support this important conclusion the authors should move the biochemical data shown in Figure 6A and B to Figure 3.

We present all the surface biotinylation studies in one figure, current Figure 6A-B. In that figure we compare the surface levels of α5-integrin, β1-integrin and LRP1. We would like to keep this order, so that the first focus on the paper is on β1-integrin, and after establishing a phenotype, we now test α5-integrin and LRP1, another SNX17 cargo.

In addition, they should supplement these surface restricted biotinylation experiments, through inclusion of a negative control, that is a cell surface cargo whose level does not decrease upon PIKfyve inhibition (this could, for example, be a retromer pathway specific cargo protein). In this way, the authors can address whether PIKfyve is causing a global perturbation of endosomal recycling (that is, its effects are not restricted to solely the SNX17-retriever-CCC-WASH pathway) or is it having a selective effect on this pathway.

We agree that it is important to determine if other receptors that traffic through endosomes are affected. We had previously published that the rate of trafficking of EGFR, which goes from the cell-surface via endosomes to lysosomes was not affected in Vac14-/- primary fibroblasts (PMID 17956977). Here, we tested and found using immunofluorescence, that the surface levels of EGFR are the same in cells treated for 1 hour with either DMSO or apilimod (Figure S6). Note also in a study of MDCK cells, Dukes et al. 2012, found that while PIKfyve inhibition resulted in some tight-junction proteins accumulating in intracellular compartments, other tight-junction proteins were not impacted.

While the authors quantify that the steady-state level of B1 integrin is not reduced upon 30 minutes of PIKfyve inhibition (Figure S4), it would be informative to extend this timeline and include experiments with cycloheximide to define if integrin turnover is enhanced. This may provide insight into whether PIKfyve activity is required for integrin retrieval from lysosomal degradation or for post-retrieval transport from the endosomal compartment.Endosomal sorting of active versus inactive forms of integrin heterodimers is an important feature of their role in cell migration. Antibodies specific for active and inactive conformation are available, and these could be incorporated into the study to provide a view of whether PIKfyve is required for the recycling of integrins independently of their activity status or is more selective for sorting of active integrins (the SNX17-retriever-CCC-WASH pathway is generally considered to support recycling of active and inactive integrin).The quantification of endosomal association of the SNX17-retriever-CCC-WASH pathway components relies on a normalized colocalisation analysis with VPS35. A key assumption is that the endosomal association of VPS35 is not affected in anyway by the drug treatments (within the context of the rather small affects being quantified).

For these experiments, we did not rely on the intensity of VPS35. Rather, we made a mask of the VPS35 puncta, and calculated the intensity of the protein of interest in those puncta. This is now explained in the Methods. “To measure the intensity of proteins on VPS35-positive endosomes, for each field of cells, a mask was created in ImageJ using VPS35 fluorescence and then overlaid onto the fluorescence channel of the protein of interest.”

This information was also added to the Figure legend of Figure 7. “A mask of the VPS35 puncta was generated in Image J, and the intensity of VPS35L, COMMD1, FAM21 and SNX17 on VPS35-positive endosomes was quantified and values were normalized to the corresponding average intensity of the DMSO treatment cohort.”

After all, a sub-population of VPS35 is know to be associated to endosomes through binding to the PI3P-binding protein SNX3, with the remaining VPS35 population being associated through RAB7-GTP – a GTPase that itself is controlled through endosomal phosphoinositides. Teasing apart the precise roles of PI3P and PI3,5P2 in the context of endosomal association is therefore extremely difficult especially when the perturbation in localization are very small. It would perhaps be more appropriate to place less emphasis on the role of these phosphoinositides in endosome recruitment (with the exception of SNX17 and COMMD's where the evidence is perhaps more clear), by discussing the possibility that the data reflect a role for the phosphoinositide switch in aiding the timing of coupling and communication within the pathway (e.g. could be through conformational changes induced by phosphoinositide binding or altered protein orientation on the membrane).

When we consider direct binding to a phosphoinositide, we assume that this could be recruitment of a protein(s), conformational changes of a protein or altered orientation of a protein on a membrane. In addition, we agree that the roles of phosphoinositides could be indirect via proteins in another pathway that bind directly, but then indirectly impact the pathway being investigated. One example would be potential roles of PIKfyve in maturation of endosomes. We have now included a discussion of this possibility in the manuscript.

“While we favor a model where positive regulation of the SNX17-Retriever-CCC-WASH pathway occurs via direct binding of some of the subunits within these complexes to PI3P and PI3,5P_2_ or PI5P, it is possible that these lipids are impacting the pathway in a more global way. For example, PIKfyve is important for endosome maturation (Kim et al., 2014; Messenger et al., 2015), and this could play a role in when the SNX17-Retriever-CCC-WASH pathway is activated.”

The generation of the COMMD1-EDEQE mutant is predicted not to change its overall structure. The authors should validate that when expressed in cells this mutant can retain association to CCDC22 and CCDC93.

We thank the reviewers for raising this question, because following this suggestion resulted in our discovery of additional problems with the COMMD1 mutants used in the original study. We now tested and found that the EDEQE mutant greatly perturbs the association between COMMD1 and CCDC93 (Author response image 1). Thus, we removed the EDEQE mutant from the manuscript.

**Author response image 1. sa2fig1:** Mutation of the putative phosphoinositide binding site on COMMD1 impairs the interaction with CCDC93. HeLa cells were infected with lenti-virus expressing 3xHA-COMMD1 or 3xHA-COMMD1-EDEQE (R120E/ K129D/R133E/H134Q/K167E) mutant for 48 hours. 3xHA-COMMD1 were immunoprecipitated with HA antibodies and the precipitates were immunoblotted for CCDC93. The binding affinity for COMMD1 mutants with CCDC93 was quantified from three independent experiments. The intensity of co-immunoprecipitated CCDC93 was normalized to relevant pull-downed COMMD1. Statistical significance was analyzed using unpaired two-tailed Student’s T-test. **** P<0.0001.#.

We also noted a different issue with the COMMD1-QAA mutant and found that it was expressed at levels that were 8.6-fold higher than wild-type. This very large increase raises the possibility that the large amounts of the COMMD1-QAA mutant combined with mutation of this site to a hydrophobic patch, could potentially mask any defects associated with a defect in binding to phosphoinositide lipids. Thus, we were unable to analyze the COMMD1-QAA mutant further (Figure S14).

The COMMD1-EQE mutant also presented some issues. While its expression was closer to wild-type COMMD1, we found that the COMMD1-EQE mutant also had an approximately 60% defect in binding to CCDC93 compared with wild-type COMMD1. When we accounted for the large increase in COMMD1-QAA levels, we found that it had a 40% defect in binding CCDC93, and that the differences between the COMMD1-QAA and COMMD1-EQE mutants were not statistically significant (Figure S14). We now explain the new data as follows:

“It is not surprising that mutation of a phosphoinositide binding site would also impair the formation of the CCC complex. Formation of the complex could potentially require association with membranes. Alternatively, this site may play a direct role in formation of the CCC complex. Thus, while it is possible that impaired association with PI3,5P_2_ underlies the defect in COMMD1-EQE binding to endosomes, and accounts for a defect in recycling, it is also possible that the defect in the COMMD1-EQE mutant is due to a partial disruption of the CCC complex that is independent of its interaction with phosphoinositide lipids.”

In addition, what happens to the localization of CCDC22 and/or CCDC93 when PIKfyve is inhibited? Robust and clear loss of endosomal localization of these proteins would be needed to support the statement (line 478) 'generation of PI3,5P2 and/or PI5P then recruits the CCC complex'.

We now tested and found that inhibition of PIKfyve decreases the endosomal localization of CCDC93 (new supplemental figure S7C-D).

The final model needs to better reflect the complexities of endosomal association of these sorting components. The WASH complex, for example, is associated to endosomes through retromer-dependent and retromer-independent mechanisms, and while the respective level of these mechanisms may vary between cell types it is incorrect to show that WASH association is solely reliant on PI3P, PI3,5P2 and/or PI5P.

We agree and have now changed to text to indicate that these lipids play a role. We made sure to acknowledge that they are not the sole mechanism for the assembly of this pathway.

Reviewer #3 (Recommendations for the authors):Suggestions for additional data/experiments:A. There is a strong focus on the recycling of integrins and the authors do show effects on the cell surface levels of integrins and the endosome to cell surface recycling of integrins (and to a lesser degree, LRP1). I think it would be helpful however if the authors could show something that doesn't change when PIKfyve is inhibited. For example, do cell surface levels of the transferrin receptor, Glut1 or EGFR change in any way when PIKfyve in inhibited? To me, this is a necessary control that will confirm that PIKfyve has a function in sorting/transport of specific ligands from endosomes to the cell surface.

We agree that it is important to determine if other receptors that traffic through endosomes are affected. We had previously published that the rate of trafficking of EGFR, which goes from the cell-surface via endosomes to lysosomes was not affected in Vac14-/- primary fibroblasts (PMID 17956977). Here, we tested and found that the surface levels of EGFR are the same in cells treated for 1 hour with either DMSO or apilimod (Figure S6).

B. Similarly, when the authors have investigated effects of the expression of the VPS34-IN1 (Figure S7) they show that there are effects on every protein they investigate. Is there an endosomal protein that is not affected by the VPS34-IN1?

While it would be optimal to find an endosomal protein that is not affected by VPS34-IN1, we currently do not have a good guess of what to test. As we show in this manuscript (Figures S9 and S11) treatment with Vps34-IN1 lowers PI3P, as well as impacting PIKfyve function which results in lowered PI3,5P_2_ and PI5P as well. Thus, we predict that many proteins would be affected. Importantly however, the interpretation that SNX17 and FAM21 are recruited in part by PI3P are based on multiple publications and are not solely based on the inhibitor studies reported here.

I would also urge the authors to employ a biochemical assay to examine membrane association as relying purely on microscopy can give somewhat limited data. For example, an assay that separates membranes from cytosol by centrifugation would be useful to corroborate/support the observations made through microscopy. Certainly, some of the effects they observe when PIKfyve is inhibited are quite marginal and not necessarily consistent with PIKfyve (and PI3,5P2) playing such a key role.

We agree that an alternate biochemical method to measure the changes in endosomal localization of the proteins would be ideal. However, we found that only about 10-15% of each Retriever, CCC and WASH subunit tested is associated with membranes (Reviewer Figure A). Thus, we did not pursue biochemical approaches further.

Given that only a small population is on membranes, using immunofluorescence to focus on changes at a specific compartment, for example, Vps35 containing membranes, provides a more sensitive assay.

Suggestions for changes to the text:A. The authors highlight the role for PI3P in mediating the recruitment of Fam21 to endosomes but there is a more substantial body of data that show that binding of Fam21 to the retromer protein VPS35 mediates recruitment of the WASH complex to endosomes (see PMIDs: 20923837; 22070227; 23331060; 22513087; 24819384). The effects of PI3 kinase inhibition on the endosomal pool of Fam21 appear fairly marginal which might indicate that binding PI3P helps stabilise Fam21 membrane association, but it cannot be concluded that Fam21 recruitment requires PI3P. In other experiments that authors make similar claims regarding the importance of PI3P or PI3,5P2 when the effects observed as rather marginal. Toning down the text to reflect the reality of the observed effect would be beneficial.

We agree with the reviewer and we have added a statement about the importance of the retromer for the endosomal association of the WASH complex. We have also toned down the text about potential roles of the phosphoinositide lipids.

B. Similarly, when the authors describe the role of retriever in endosome to cell surface recycling, they state that half of the cell surface proteins are trafficked by this route but what do they mean by 'half'. Are half of the many species of cell surface proteins affected by loss of retriever function or is there literally half the number of cell surface protein present at the cell surface after loss of retriever?

We agree and now make a more neutral statement that “Proteomic studies revealed that this pathway traffics over 120 cell surface proteins in HeLa cells (McNally et al., 2017), which establishes SNX17-Retriever-CCC-WASH as an important pathway for protein recycling.”

C. The authors should recognise (and cite) literature showing that Snx27 may also mediate endosome to cell surface recycling of integrins (see PMIDs: 27909246; 23563491). Indeed, levels of Retriever component expression (except VPS29) are often much lower than levels of retromer components suggesting that retriever may sort fewer membrane proteins (see PMID: 27278775) and that therefore, some membrane proteins could be sorted by both pathways.

We agree that these studies leave open the possibility that SNX27 may also mediate endosome to cell surface recycling of integrins. We now include this point and the three references listed above in the introduction.

[Editors' note: further revisions were suggested prior to acceptance, as described below.]

Reviewer 1 wrote, "I was disappointed by the lack of effort to biochemically verify the observed changes in membrane localization of SNX17 and retriever components. The authors state that only 15% of SNX17 and retriever components are membrane bound, which is plausible. Yet, the 15% left on membranes after digitonin permeabilization result in a good western blot signal in the panel that they provide. Please quantify the blot and add the data.

We have now quantified the blot and include this information in new Figure 7—figure supplement 6.

Reviewer #1 (Recommendations for the authors):The authors have performed substantial new experimentation to address my concerns regarding integrin surface localization. They have also confirmed their inhibitor based results with genetic perturbation of PIKfyve, which strengthens the study.However, I was disappointed by the lack of effort to biochemically verify the observed changes in membrane localization of SNX17 and retriever components. The authors state that only 15% of SNX17 and retriever components are membrane bound, which is plausible. Yet, the 15% left on membranes after digitonin permeabilization result in a good western blot signal in the panel that they provide. Please quantify changes in this membrane bound fraction upon Apilimod treatment. It is a key result in their study and two reviewers suggested (even "urged", see reviewer 3) that the microscopy should be confirmed by a biochemical method. The method seems to work and there are changes upon Apilimod treatment. It is a simple assay, the signal is robust enough to be quantified and the microscopy data alone is not convincing enough.I think that the authors should repeat this fractionation a few times to quantify the changes in membrane bound SNX17 and retriever components. If this confirms their microscopy, their model is likely correct and the study publishable. If it does not confirm the microscopy, the model is probably wrong.

We agree that ideally there should be an orthogonal biochemical approach. However, the biochemical assay that we tested, mild digitonin treatment, extracted more than 81% and 95% of SNX17 and COMMD1 respectively. It is not possible to obtain good quantitative data of an additional 15-20% lowering of this small amount of remaining protein. A better in vitro assay needs to be developed, however this is outside the scope of this manuscript.

Reviewer #3 (Recommendations for the authors):The authors have made a fair effort to address the concerns I raised previously. Whilst some issues remain unresolved, I recognize that this is likely to be due to technical challenges rather than a lack of willingness to make the necessary revisions.Overall, I think this study has merit and will be of interest to the field. I'm not sure it is fundamentally 'game-changing' but that should not be a barrier to publication.

Thank you for the recognition of the technical challenges, and the new findings reported here.